

# Interannual variability in air temperature and snow drive differences in ice formation and growth

Arash Rafat[1] and Homa Kheyrollah Pour[1]

[1]Cold Regions Research Centre and Department of Geography and Environmental Studies, Wilfrid Laurier
University, Waterloo, ON, Canada,

*Correspondence to*: Arash Rafat (arafat@wlu.ca)

**Abstract**

Recent warming of northern, high-latitude regions has raised alarms for the safe and efficient use of frozen lakes for winter transportation and recreation. This concern is significant in Canada's Northwest Territories (NWT), where

seasonally constructed roads over lakes, rivers, and land (winter roads) span thousands of kilometers and act as vital links to isolated communities and resource development projects. Current climate change and weather variability is altering the evolution of lake ice, challenging predictions of freeze-up, ice growth, and ice decay. The accurate simulation of ice evolution is imperative for safe and efficient planning, operation, and maintenance of winter roads under a changing climate and heightened weather variability. This is particularly significant in the early winter

period when ice road planning and design is undertaken. Here, we investigate the effects of weather variability on ice formation, growth, and evolution in a small lake near Yellowknife, NWT, Canada. High-resolution measurements of air, snow, ice, and water temperatures were collected continuously from a floating research station between October and December in 2021, 2022, and 2023 and variability in ice evolution and weather examined. Combinations of above and below average snowfall and winter air temperatures resulted in variability of up to 17

days in freeze-up dates (FUD) and 8 days in freeze-up durations. End of December ice thicknesses ($h_i$) varied up to 12 cm, while the duration between the FUD and $h_i$=30cm varied up to 10 days. $h_i$ were effectively simulated (RMSE=1.11-2.33 cm) using empirical relationships developed using cumulative freezing degree days ($CFDD$) and seasonally cumulative snowfall ($S_T$), while snow-ice thicknesses simulated (RMSE=0.83-1.21 cm) using $CFDD$ and daily snowfall. Developed relationships between air temperatures, snow, and ice thicknesses can be used for

predicting minimum ice thicknesses required for commencing ice road construction, and to assist in the effective management of construction activities.



## 1. Background

Ice covers act as critical infrastructure for northern regions by means of seasonally constructed roads over lakes, rivers, and land (winter roads). Winter roads allow for the cost-effective transportation of vital goods and services to isolated communities (Barrette et al., 2022) and remote mining projects (Hayley and Proskin, 2008). Winter roads support critical resource development projects across Canada and contribute substantially to the Canadian economy (Prowse et al., 2009). Further, the presence of snow and ice over lakes inherently affects lake ecological and geochemical processes (*e.gs*. Huang et al., 2021; Song et al., 2019) and are used as indicators of climate change (Kheyrollah Pour et al., 2014a; Kheyrollah Pour et al., 2014b; Palecki & Barry, 1986; Skinner, 1993).

Phenological changes in lake ice covers have been explored across many northern, high-latitude regions and strongly relate to weather conditions (Huang et al., 2019; Latifovic and Pouliot, 2007; Leppäranta et al., 2017). There is coherence amongst most published literature that lakes across the northern hemisphere are experiencing earlier break-up dates (BUDs), with some exceptions depending on time periods analysed, significance levels attributed to trends, and specific regions. Duguay et al. (2006) report earlier BUDs in 81 lakes across Canada between 1961-1990. L'Abée-Lund et al., (2021) observed similar trends in 101 lakes in Norway between 1870-2010 as did Hallerbäck et al. (2022) in 47 lakes in Sweden. The trend extends to 18 lakes Poland (1961-2010; Choiński et al., 2015), Lake Baikal (1869-1996; Todd & Mackay, 2003)), and the 65 lake lakes in the Laurentian Great Lakes Region (1975–2004, Jensen et al. 2007). Meta-analysed conducted by Newton and Mullan (2021) for 678 lakes spread mostly across North America and northern Europe show similar results, as do studies derived from the Global Lake and River Ice Phenology (GLRIP) Dataset produced by Benson et al. (2002).

Trends in freeze-up dates (FUDs) have shown much greater spatial variability, as ice formation depends strongly on local topography, lake morphology, and lake heat storage (Leppäranta, 2015) which are not commonly reported. Regional trends in FUDs are often masked out, under-represented, or are not available, particularly in meta-analyses where a majority of lake may show later FUDs. For instance, Sharma et al. (2021) emphasize a global trend toward later FUDs, yet, in their study of 60 lakes spread across 8 countries, only 31 lakes had viable data for FUDs, of which, the length of data records, definitions used for delineating FUD, and method of observation of ice formation vary in space and with time. The definitions used for delineating ice phenological events are significant, and are a challenge in accruate time-series analysis of ice phenology (Catchpole and Moodie, 1974; Wynne, 2000). The length of available data record has also been noted to the estimated magnitudes of trends (Benson et al., 2012; Supplementary Material; Sharma et al., 2021; Supplementary Material ). Several notable examples of lakes with earlier observed trends in FUDs can be found in the published literature including in Finnish Lapland (1930s-1960s; Korhonen, 2006), Xinjing (2001-2018; Cai et al., 2020), eastern Canada (1961-1990) and the Great Lakes-St. Lawrence regions (1951-1980) (Duguay et al., 2006), Kazakistan and Tajikistan (2002-2022; Hou et al., 2022), Lativa (1945-2002; Apsīte et al. 2014), Poland (1960-1989; Girjatowicz et al., 2022), Sweden (1913-2014; Hallerbäck et al., 2022), and in the Qinghai-Tibetian Plateau (2002-2021; Sun et al., 2023; 2000-2011; Yao et al., 2016). Trends of earlier FUDs in the last 30 years are of particular interest as they largely contrast findings presented in Newton and Mullan (2021) and Sharma et al. (2021) who argue synchronicity in later freeze-up dates.



Recently, Basu et al. (2024) extended on Sharma et al. (2021) using ice-on dates from 1,899 lakes between 1971-2020, mostly in North America and Europe, and found similar trends of earlier BUDs and later FUDs to those in the predecessor study. It is reiterated here that differences in methods, satistical analyses and time-periods of analysis directly contribute to variences in trends of both BUDs and FUDs regardless of the conducted study is for a singular site or via meta-analyses.

To better understand interactions between weather and climate and ice formation and growth, high-frequency, in-situ observations of interactions between air, snow, ice, and water should be monitored. Doing so allows for the development of physics-based and/or empirically derived understandings of ice-atmosphere interactions for integration in lake models and global upscaling. In particular, improved understanding of ice formation and growth in early winter (October-December) can be essential for the effective and safe scheduling of the operating windows, choice of construction equipment, and the hazard control plans for ice road design.

In this study, we investigate the effects of inter- and intra- annual variability on ice formation and growth in a small subarctic lake in the Northwest Territories (NWT), Canada. We compare weather variability in three early winter periods (September-December 2021, 2022, and 2023) against the historical climate record (1942-2023) to understand driving forces in variability in freeze-up, ice-onset, and ice growth. Empirical relationships between snow-ice, total ice, and snowfall are developed using the acquired data for practical consideration for ice road design process.

## 2. Study area and floating research station

This study aims to relate weather variability with ice formation and growth within Landing Lake, a small subarctic lake ~11 km north of Yellowknife, NWT, Canada (Fig. 1). Landing Lake has a surface area of 1.07 km$^2$, and mean and maximum depths of 1.77 m and 4.28 m respectively (Rafat et al., 2023). The lake drains a relatively large catchment of 135 km$^2$ (Spence and Hedstrom, 2018) and is part of the larger Baker Creek Research Watershed (BCRW). BCRW is a well-studied, 155 km$^2$ Canadian Shield subarctic watershed consisting of 349 small lakes (Spence and Hedstrom, 2018). The watershed is drained by Baker Creek which flows into Great Slave Lake. The basin is located within a region of discontinuous permafrost and large changes in topography, vegetation, hydrology, and surficial geology (Morse and Wolfe, 2017; Phillips et al., 2011). Land coverage in the BCRW is split between exposed bedrock (~40%), water (~22.6%), forested hillslopes (~21.5%), and wetlands/peatlands (~15.9%) (Spence and Hedstrom, 2018).

Yellowknife has a subarctic continental climate (Koppen Dfc). Annual precipitation is 288.6 mm, with 157.6 cm of snowfall. Snow begins to accumulate in October and melts in April. Climate normal (1981–2010) mean annual air temperatures are −4.3°C. Summers are cool with peak mean daily air temperatures in July of 17.0°C. Winters are cold and dry. Below freezing air temperatures persist for > 6 months of the year, with January mean daily air temperatures of −25.6°C.





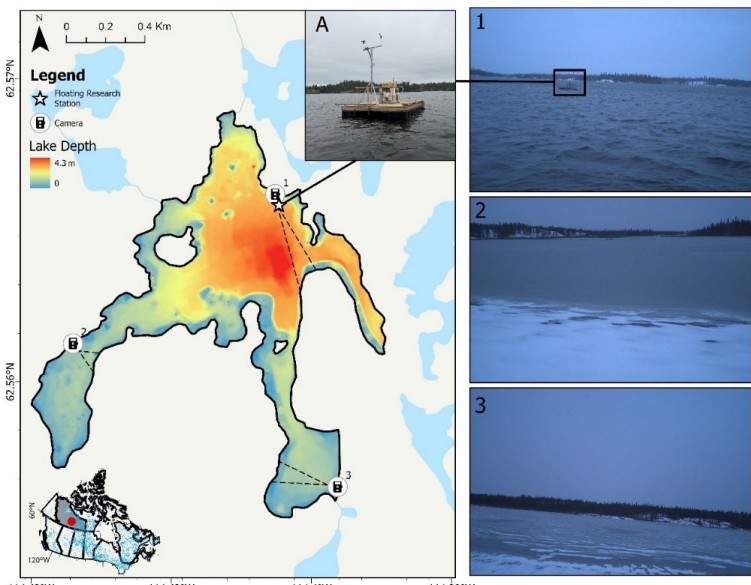

**Figure 1: Site map of Landing Lake, Northwest Territories, including photographs of the Floating Research Station, and**
**perspectives from trail cameras (1, 2, and 3). Photographs 1, 2, and 3 were taken on October 23, 2023, at 09:00 local time.**

In October 2022, a Floating Research Station (FRS) was built and anchored in Landing Lake to monitor the annual
evolution of ice and snow. The FRS was anchored at a location 3.00 m deep (62.56 N, 114.40 W) and consisted of a
2.44 x 2.44m floating structure. Instrumentation attached to the FRS included a Snow and Ice Mass Balance
Apparatus (SIMBA) thermistor chain, 2 pressure transducers (1 in water, 1 in air), 10 digital temperature sensors
near the lakebed, a CTD sensor (YSI EXO2 Sonde), two photosynthetically active radiation (PAR) sensors within
the water, and a weather station tripod which measures wind speeds and directions, air temperature, relative
humidity, and incoming and outgoing shortwave radiation. PAR sensors were installed within the water column at
heights 0.85 m and 1.27 m above the sediments. A larger meteorological station located 100 m east of the raft on an
island in Landing Lake provides supplemental measurements of turbulent and radiative heat fluxes (Spence and
Hedstrom, 2018). Figure 2 presents a schematic of the FRS.





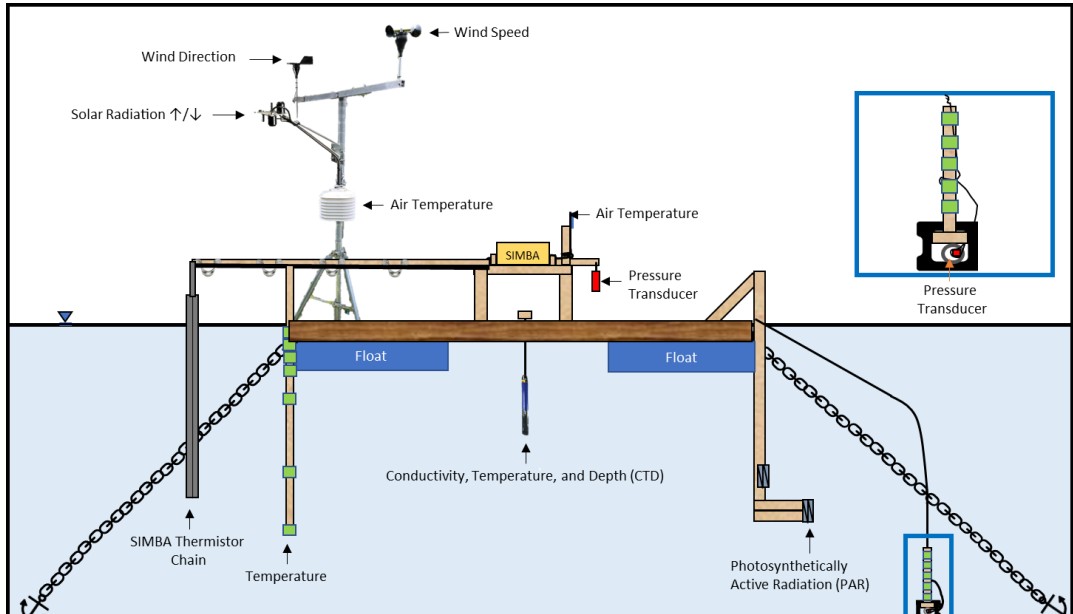

Figure 2: Instrumentation onboard the Floating Research Station, Landing Lake, Northwest Territories.

## 3. Methodology

### 3.1. Interface detection and manual observations

Air, snow, ice, and water interfaces were identified using the SIMBA. The SIMBA operated in two modes. Mode I took direct measurements of the ambient temperatures surrounding each sensor every 15 minutes. Mode II operated every 6 hours. Over a 2-minute heating cycle, Mode II applied a 64mW constant and linear heat source to resistors housed beside each of the 145 sensors. The associated temperature rise at each resistor was recorded by adjacent

temperature sensors every 30 seconds. Four (4) additional measurements were taken in the proceeding 2 minutes to measure the cooling response at each resistor as the applied power is stopped. Mode II provided a means of approximating the thermal conductivity ice and snow, mimicking the transient hot-wire method for measuring thermal conductivity (Healy et al., 1976; ASTM D5334) and allowed for improved interface detection (Jackson et al., 2013). The SIMBA has been widely used in monitoring sea ice (e.gs. Koo et al., 2021; Lei et al., 2018), and

more recently in river ice (Lynch et al., 2021), and lake ice evolution (Cheng et al., 2021; Rafat et al., 2023).

The position of the ice (or water) surface was identified by combining the information from Mode I and Mode II of the SIMBA. The temperature rise after 2 minutes of gentle heating was lower in water compared to air, as water has a larger heat capacity. Likewise, the temperature rise in ice was lower as compared to air, given ice's comparably

large thermal conductivity and density, thereby effectively transferring heat away from the source. As compared to one another, however, the temperature rise in water, ice, and slush are similar, normally ranging between 0.4°C and 0.75°C. Snow is excluded from this range as temperature rises in snow are significantly higher. Therefore, beginning



at the top of the chain, the position of either the water, ice, or slush surface would be the first sensor where the temperature rise after 2 minutes of heating would be between 0.4°C and 0.75°C. To delineate between water and ice,
Mode I measurements at the identified location were analyzed. If the temperature ($T$) was ≤ - 0.5°C, the surface was identified as ice, if $T ≥ 0.125$°C the surface was water, and if -0.5°C < $T$ < 0.125°C, the surface was unfrozen slush.

The position of the air-snow interface was identified by selecting the location where the spatial derivative was a maximum, beginning from the top of the thermistor chain. Since snow is an effective insulator, vertical gradients in
temperatures would present a distinct peak when transitioning from air to snow. To identify the position of the ice bottom, Mode I of the SIMBA was used. For each measurement, the thermistor chain was searched between the bottom of the chain and the identified ice (or water) surface. The first sensor with -0.5°C ≤ $T$ ≤ -0.0625°C was selected to be position of the ice bottom, provided that the sensor immediately above this identified location also fell within the noted range to reduce uncertainty. The range was selected to account for the manufacturer reported
accuracies of the sensors and the minimum resolution of the thermistors of ±0.0625°C. Further details on interface detection and its validation in Landing Lake are presented in Rafat et al. (2023).

### 3.1.1. Manual and pre-FRS observations

No direct measurements of air, surface, or water temperatures were available prior to the first installation of the SIMBA in Landing Lake on December 6th, 2021. For determining the freeze-up date (FUD) and date of ice-onset
(IO), MODIS-derived surface temperatures and Sentinel-2 optical imagery were used. FUDs measured in-situ at the FRS in 2022 and 2023 were verified using the same approach for validation. Imagery was acquired from Sentinel Hub's EO browser (https://www.sentinel-hub.com/). In this study, we defined the freeze-up duration as the period between the first occurrence of ice (ice-onset; IO), and the formation of a solid ice cover over the entire lake (FUD). Manual measurements of snow depth, and ice thickness were taken between November-December 2021-2023 for
comparison with the SIMBA measurements. Three trail cameras (RECONYX Hyperfire 2) were installed along the shoreline of Landing Lake to capture the freeze-up process. One camera was installed in October 2022 (Camera 1, Fig. 1) while two more cameras were installed in October 2023 (Cameras 2 and 3, Fig. 1).

### 3.2. Air and snow parameters and frequency analysis

To understand the influence of weather variability on ice growth, air temperature and snowfall parameters between
September-December 2021, 2022, and 2023 were compared with in-situ measurements of ice and snow evolution. Several air temperature and snowfall parameters were selected in this study for achieving this objective. For air temperatures, daily mean air temperatures ($T_m$) measured from the Yellowknife Airport climate station were chosen. ($T_m$ are averaged for each month between September-December, and for the bulk September-December 4 month period. As a first-order approximation of ice growth potential, the cumulative freezing degree days ($CFDD$)
in each month and for the September-December period were calculated. $FDD$ is ubiquitously used for estimating ice growth using Stefan's equation. For evaluating interannual variability in snowfall, several parameters were selecting including the day of which the first snowfall occurred ($S_{ON}$), the cumulative snowfall ($S_T$), the peak hourly snowfall





rate in a given day in each month ($S_p$), and the number of snowfall days ($S_d$). Snowfall was recorded at the Yellowknife Airport weather station. In this study, we define the timing of zero-degree isotherm as the first date

when mean daily air temperatures fell and remained below freezing (0°C) for 3 consecutive days.

The severity of a given snowfall parameter ($S_T$, $S_{ON}$, $S_p$, and $S_d$) was evaluated using a frequency analysis conducted on the entire observational data record from the Yellowknife Airport weather station (1942-2023). For a given parameter, the probability of exceedance ($P_e$) in any given year was determined using Eq. (1), where $m$ is rank of the data, and $n$ is the length of the data record (82 years). The return period ($R_P$) was determined using Eq. (2).

$$P_e = \frac{m}{n+1} \tag{1}$$

$$R_P = \frac{1}{P_e} \tag{2}$$

A Log-Pearson Type III (LP3) distribution was fit to the ranked data. LP3 is a well-studied distribution commonly used in hydrological applications and has been extensively used in flood frequency analysis and forecasting (Bobée, 1975). The logarithm of each parameter $Y$ was determined using Eqs. (3a-c). The antilog of $Y$ values are evaluated following Eq. (3c) for interpretability.

$$K = \frac{2}{G}\left\{ \left( \left[ z - \frac{G}{6} \right] \frac{G}{6} + 1 \right)^3 - 1 \right\} \tag{3a}$$

$$G = \frac{n}{(n-1)(n-2)} \sum_{i=1}^{n} \left( \frac{\log(Y_i) - \log(\overline{Y_i})}{\sigma_l} \right)^3 \tag{3b}$$

$$Log(Y) = \overline{\log(Y)} + K\sigma_l \tag{3c}$$

Where $G$ is the skewness coefficient, $K$ is the frequency factor depending on the return period and skewness, $n$ is the length of record, $\overline{\log(Y)}$ and $\sigma_l$ are the mean and standard deviation of the logarithm of snowfall totals for any given month over the entire data record, and $z$ taken as the standard normal deviate.

### 3.3. Heat storage

Freeze-up is directly related to the heat storage within a lake. Hence, it is necessary to estimate the heat storage
withing the water column of a lake to understanding the ice freeze-up and growth process. The rate of change in heat storage ($\dot{E}_T$) within Landing Lake was estimated using measurements of temperature at each sensor along SIMBA's thermistor chain located on the FRS (Eq. 4). $h_{i,btm}$ represents the ice bottom (or water surface if no ice has formed), $z_T$ the lowest measurement points along the thermistor chain, $\rho_w$=1000 kg m$^{-3}$ and $c_{p,w}$ =4186 J Kg$^{-1}$ °C$^{-1}$ represent the density and heat capacity of freshwater, and $T_w$ represents the SIMBA-measured water temperature. $\dot{E}_T$ is
presented in units W m$^{-2}$.



$$\dot{E}_T = \frac{\partial}{\partial t} \int_{h_{i,btm}}^{z_T} \rho_w c_{p,w} T_w(z) dz \qquad (4)$$

## 4. Variability in weather and climate

Mean daily air temperatures and cumulative snowfall between September-December 2021, 2022, and 2023 displayed large interannual variability and variability in reference to the climate normal and preceding 30-year periods (Fig. 3, Table 1). $T_M$ in September, October, and November 2021-2023 were significantly greater than the 1981-2010 climate normal and 1992-2021 periods (Table 1). Both December 2021 and 2022 were colder than the climate normal and 1981-2010 periods by >3°C while December 2023 had anomalously high $T_M$=-13.6°C, being ~8°C warmer on average than both reference periods. October 2021, 2022, and 2023 $T_M$ had above freezing temperature in contrast to both the climate normal and preceding 30-year record where $T_M$<0°C (Table 1). 2021, 2022, and 2023 also showed notably variability in minimum ($T_{Min}$) and maximum ($T_{Max}$) daily air temperatures, particularily in October 2021 where $T_{Min}$ >0°C and $T_{Max}$ being 3.3°C and 3.9°C greater than the climate normal and the preceding 30-year record respectively. With $T_{Min}$ >0, even during calm nights, ice growth potential is dramatically hindered. Despite interannual and long-term variability in $T_{Min}$ and $T_{Max}$, the range between, $T_{Max}$-$T_{Min}$, remained similar in reference to the climate normal and preceding 30-year periods. This suggests that diurnal variability in weather was not significantly different between the 2021- 2023 and the two reference periods on average over the months.

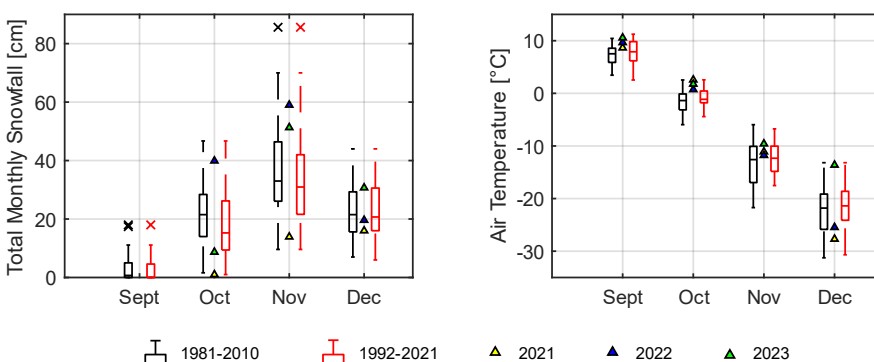

**Figure 3: Comparison of a) cumulative monthly snowfall and b) mean daily air temperatures for the September-December period in 2021, 2022, and 2023 against the climate normal (1981-2010) and preceding 30-year record (1992-2021) periods.**

Except for December 2021 and 2022, $CFDD$ in all months between September-December 2021-2023 were lower than normal, indicating warmer than normal conditions. October 2021 was particularly warm (and dry) having $CFDD$ =7.9°C Day (Table 1). The same year saw colder than normal conditions by the end of December with $CFDD$ being 123% of normal. In 2023, conditions were exceptionally warm, resulting in end of December $CFDD$ reaching only 65% of normal.



Snowfall between September-December 2021-2023 was highly variable (Fig. 3, Table 1). October 2021 had nearly no snow (1 cm), while October 2022 and 2023 had 39.9 cm and 8.7 cm respectively. $S_T$ by the end of October 2021 was only 4% of normal while October 2022 was 186%. Similar variability was recorded in November 2021, 2022, and 2023 having 13.9, 59.0, and 51.3 cm (38%, 160%, and 139% of normal) respectively. 2021 was dry, with only 30.9 cm of snowfall falling over the entire September-December period, as compared to 85.1 and 77.3 cm for the climate normal and 1992-2021 periods, respectively. This resulted in the second-lowest recorded October $S_T$, surpassed only by October 1944, which had a total recorded snowfall of 26.1 cm. Between September-December 2022, $S_T$ was 139% and 153% of the climate normal and preceding 30-year period respectively. In 2023, these values decreased to 107% and 117% respectively. On a monthly basis, October and November of 2022 had the largest $S_T$, being 186% and 160% of their respective climate normal.

**Table 1. Comparison of air temperatures and snowfall between 2021, 2022, and the 1981-2010 climate normal**

| | Air Temperatures: $T_M$ ($T_{Min}$- $T_{Max}$) [°C] | | | | |
|---|---|---|---|---|---|
| Month | Climate Normal (1981-2010) | 1992-2021 | 2021 | 2022 | 2023 |
| Sept. | 7.2 (4.1,10.5) | 7.7 (4.3, 11.1) | 8.7 (5.1, 12.1) | 9.7 (4.7, 14.5) | 10.6 (6.7, 14.4) |
| Oct. | -1.7 (-4.0, 1.1) | -0.80 (-3.3, 1.7) | 2.6 (0.1, 5.0) | 0.72 (-2.9, 4.4) | 1.8 (-1.1, 4.7) |
| Nov. | -13.7 (-17.1, -9.7) | -12.4 (-16.0, -8.8) | -11.1 (-14.7, -7.5) | -11.8 (-15.0, -8.5) | -9.6 (-13.2, -6.0) |
| Dec. | -21.8 (-25.8, -18.1) | -21.7 (-25.4, -17.9) | -27.7 (-31.3, -24.1) | -25.5 (-29.3, -21.6) | -13.6 (-18.1, -9.1) |
| Sept.-Dec. | -7.4 (-15.3, -6.7) | -6.8 (-10.0, -3.7) | -6.9 | -6.7 | -2.7 |
| | Cumulative Freezing Degree Day [°C·day] | | | | |
| Month | Climate Normal (1981-2010) | 1992-2021 | 2021 | 2022 | 2023 |
| Sept. | 1.6 | 1.6 | 0 | 0 | 0 |
| Oct. | 75.4 | 68.7 | 7.9 | 57 | 61.4 |
| Nov. | 373.2 | 367.1 | 333.9 | 353.6 | 289.1 |
| Dec. | 654.5 | 656.3 | 803.2 | 764.2 | 422.4 |
| | Snowfall [cm] | | | | |
| Month | Climate Normal (1981-2010) * | 1992-2021 | 2021 | 2022 | 2023 |
| Sept. | 3.6 (0, 18.0) | 2.6 (0.0, 18.0) | 0 | 0 | 0 |
| Oct. | 21.4 (1.6, 46.7) | 18.0 (1.0, 46.7) | 1 | 39.9 | 8.7 |
| Nov. | 36.9 (9.6, 85.6) | 33.6 (9.6, 85.6) | 13.9 | 59.0 | 51.3 |
| Dec. | 23.2 (7.0, 44.0) | 23.1 (6.0, 44.0) | 16 | 19.6 | 30.7 |
| Sept-Dec. | 85.0 (52.0, 128.7) | 77.2 (30.9, 128.7) | 30.9 | 118.5 | 90.7 |

*mean (min, max) monthly cumulative snowfall between 1981-2010*





Interannual variability in total monthly snowfall can be decomposed into variability in total number of snowfall days

235    per month, $S_d$, and the maximum daily snowfall magnitude in each month, $S_p$ (Table 2). Both parameters can be

contextualized for each season by evaluating the timing of the first snowfall of a given winter ($S_{ON}$). Low $S_T$ during

the September-December period in 2021 was explained by a combination of lower than normal $S_n$ (11 days) and $S_P$

(0-2.8 cm hr[-1]) in all months. By the end of December in 2022, $S_T$ was higher than normal, which can be

explained by higher than normal $S_P$, as $S_d$ remained near normal and $S_{ON}$ was 17 later than normal (Table 2).

Although $S_n$ was near-normal, when considering $S_d$ in the context of the number of days between $S_{ON}$ and

December 31, 2022 (77), nearly 1 in 2 (37/77; 48%) days had recorded snow. This is compared to snowfall

occurring 2 in every 5 days (40%) for the climate normal period.

**Table 2: Interannual Variability in daily snowfall, the number of snowfall days in each month, and the timing of the first**
**winter snowfall.**

|  | $S_p$ [cm d[-1]] | | | | | $S_d$ [days] | | | | |
|---|---|---|---|---|---|---|---|---|---|---|
|  | 1981-2010 | 1992-2023 | 2021 | 2022 | 2023 | 1981-2010 | 1992-2023 | 2021 | 2022 | 2023 |
| $S_{ON}$ [DOY] | 272 | 276 | 279 | 289 | 279 | - | - | - | - | - |
| Sept. | 2.0 | 1.0 | 0.0 | 0.0 | 0.0 | 1 | 1 | 0 | 0 | 0 |
| Oct. | 7.0 | 6.0 | 1.0 | 13.0 | 3.4 | 9 | 9 | 1 | 9 | 7 |
| Nov. | 8.0 | 8.0 | 2.8 | 9.6 | 9.0 | 16 | 15 | 11 | 18 | 18 |
| Dec. | 6.0 | 5.0 | 2.6 | 6.2 | 6.6 | 12 | 13 | 15 | 10 | 14 |
| Sept.-Dec. | - | - | - | - | - | 38 | 38 | 27 | 37 | 39 |

When evaluating the severity of the snowfall parameters against the historical record, most parameters have return

periods of less than 10 years (Fig. 4; Table 3). A notable exception is for the total monthly snowfall in November

2022 where a return period of 21 years was estimated. Figure 4 plots fitted LP3 distributions for each of the snowfall

parameters for October, November, and December. September snowfall parameters were neglected as snowfall in

September is on average insignificant.

LP3 distributions showed excellent fits to observed data for all snowfall parameters. Errors generally increased with

increasing $R_P$. For $S_T$, root mean square errors ($RMSE$) ranged 1.91- 3.00 cm. $S_P$ had relatively low $RMSE$ of only

0.38-1.10 cm d[-1] but increased for $R_P > 10$ years where less data existed. Fitted distributions in all months, when

averaged, had accuracies in $S_d$ values within 5 days of observations. However, there is large variability between

255    years resulting in the assignment of multiple return periods for the same number of snowfall days (Fig. 4b). The first

day of recorded snowfall in any given winter was accurately represented using the LP3 distribution with an $RMSE$

of 1.50 days.




**Table 3: Return period for snowfall parameters recorded in 2021, 2022, and 2023 as compared to the historical record (1942-2023).**

| | Return Period (Yr) | | | | | | | | |
|---|---|---|---|---|---|---|---|---|---|
| | **2021** | | | **2022** | | | **2023** | | |
| | **Oct** | **Nov** | **Dec** | **Oct** | **Nov** | **Dec** | **Oct** | **Nov** | **Dec** |
| $S_T$ | 1.0 | 1.1 | 1.5 | 7.5 | 21 | 2.1 | 1.3 | 8.3 | 1.1 |
| $S_d$ | 1.0 | 1.2 | 2.4 | 2.0 | 3.8 | 1.3 | 1.5 | 3.6 | 1.8 |
| $S_p$ | 1.1 | 1.0 | 1.2 | 6.9 | 3.8 | 2.9 | 1.4 | 3.0 | 3.5 |
| $S_{ON}$ | | 3.2 | | | 9.2 | | | 3.1 | |

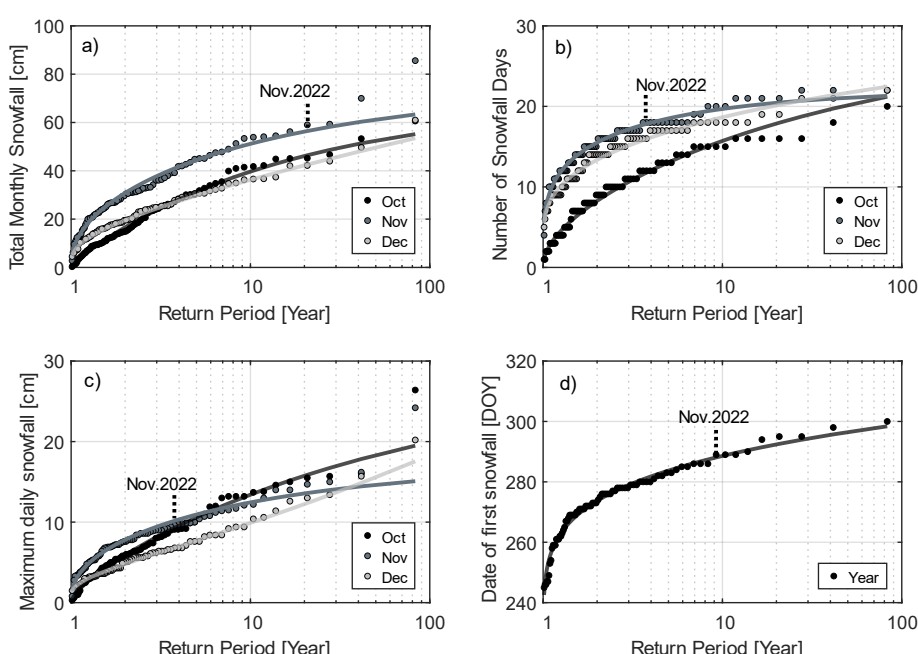

**Figure 4: Frequency analysis of snowfall parameters. a) Total monthly snowfall, b) number of snowfall days, c) maximum daily snowfall magnitude, and d) timing of the first snowfall of the year. Log-Pearson Type 3 distributions (lines) are fit to observations (circles). Each circle represents observations in any given year for October, November, and December respectively.**

## 5. Variability in ice formation and growth

### 5.1. Freeze-up

The timings of ice-onset (IO) and freeze-up dates (FUDs) between 2021-2023 were highly variable (Table 4). Between 2021 and 2022, IO varied by 20 days while FUDs varied by 15-17 days. In 2021, IO was estimated to occur on November 1 with the FUD occurring between November 7-9th, 2021. $T_a$, recorded at the Yellowknife Airport, first fell below freezing overnight on October 8. Diurnally variability above and below 0°C in hourly $T_a$ are



thought to have produced significant cooling of the lake water, which was expected to have had warmer than normal water temperatures based on above average September and October $T_a$. MODIS-derived surface temperatures during October 2021 supported this claim, ranging between 1.7-8.8°C between October 3-23. A notable cooling event likely occurred on October 18 when hourly air temperatures reached a low of -5.4°C. Despite frequent sub-freezing

temperatures, , mean daily $T_a$ remained >0°C with the exception of October 11 (-0.6 C) and October 18 (-1.1°C), until the zero-degree isotherm was crossed on October 29th (Fig. 5a). A lag time of 3 days was observed between the crossing of the zero-degree isotherm and IO. The lag time extended to ~10-13 days for the FUD with an uncertainty of 2-3 days.

In 2022, IO occurred on October 12 (Fig 5b). Diurnally oscillating air temperatures above and below freezing from

October 12-21 caused a series of freeze-melt cycles over 11 days culminating in a FUD of October 23, 2022, 3 days following the crossing of the zero-degree isotherm. Lake surface temperatures closely followed changes in air temperatures until the FUD where the surface remained slightly below freezing, while air temperatures varied between -27.5 and -0.1°C. The freeze-up duration was 11 days. In 2023, no freeze-melt cycles were recorded prior to the FUD (Fig. 5c). Air temperatures reached slightly below freezing on only two occasions before crossing the

zero-degree isotherm on October 21, 2022: Oct. 6th 01:30 (-0.38 °C) and Oct. 17th 02:30 (-0.62°C). Ice-onset was coincident with the crossing of the zero-degree isotherm on October 21, 2023. Note that in 2022 and 2023 air temperatures presented here were measured directly by the SIMBAs but in 2021 air temperatures were measured at the Yellowknife Airport Weather Station. A summary of IO dates, FUDs, and freeze-up durations in 2021, 2022, and 2023 are presented in Table 4.

**Table 4 : Interannual variability in freeze-up duration, date, and ice-onset**

|  | Timing of Zero-degree isotherm (DOY) | Ice-onset (DOY) | Freeze-up Date (DOY) | Freeze-up Duration (Days) |
|---|---|---|---|---|
| **2021** | Oct. 29[+] (302) | Nov. 1[*] (305) | Nov. 7-9[*] (311-313) | 7-9[*] |
| **2022** | Oct. 21 (294) | Oct. 12 (285) | Oct. 23 (296) | 11 |
| **2023** | Oct. 21 (294) | Oct. 21 (294) | Oct. 24 (297) | 3 |

[+]Measured by the Yellowknife Airport Meteorological Station
[*]Estimated ice onset and freeze-up dates based on Sentinel 2 imagery and MODIS-derived surface temperatures

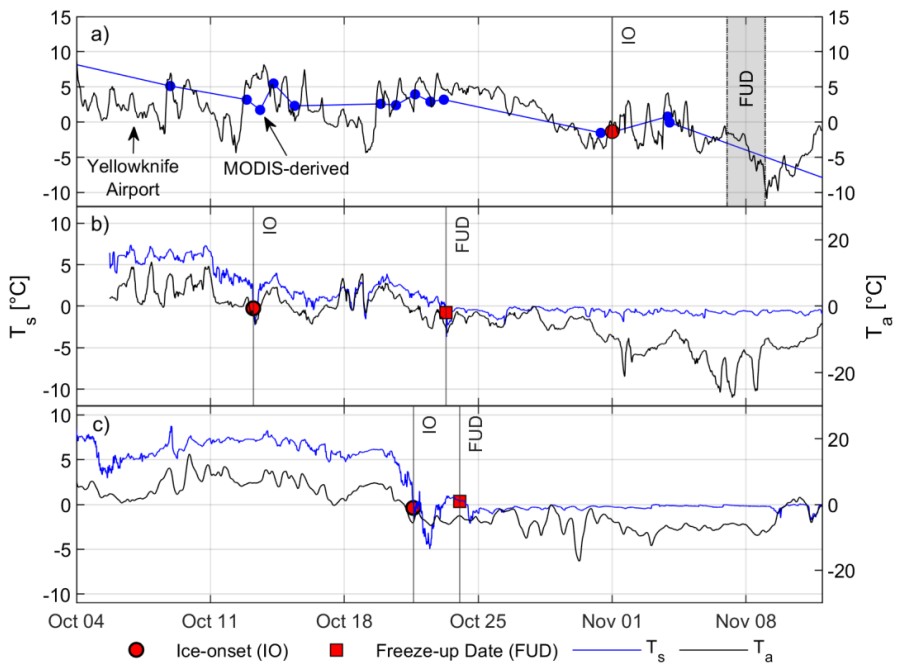

**Figure 5: Interannual variability in ice-onset (IO) and freeze-up dates (FUD) between a) 2021, b) 2022, c) 2023. Air temperatures ($T_a$) and surface temperatures ($T_s$) measured by SIMBA are presented as black and blue lines respectively. Note that in a) MODIS-derived $T_s$ and air temperatures obtained from the Yellowknife Airport weather station are presented as no SIMBA was installed prior to December 6, 2021. The grey window in a) represents a 2-day uncertainty in the FUD in 2021.**

The zero-degree isotherm was crossed on the same date in 2022 and 2023 (October 21) yet, IO dates differed by 9 days. Mean daily air temperatures in the first 2 weeks of October 2023 were significantly lower than in 2022 (Fig. 6a). Mean water ($\overline{T_w}$) and surface temperatures remained similar until October 12. Between October 4-11 differences in $\overline{T_w}$ between 2022 and 2023 ($\Delta\overline{T_w}$) were < 0.9°C (Fig. 6b). Changes in heat storage ($\dot{E}$) varied between -320.9 to 322.6 W m$^{-2}$ in 2022 and -442.4 to 249.0 W m$^{-2}$ (unsmoothed) in 2023. Between October 10-18, $\Delta\overline{T_w}$ began to diverge significantly as $\overline{T_w}$ in 2023 remained high (6.00-7.60 °C), slightly warming between October 10-14. $\overline{T_w}$ in 2022 declined beginning October 10 at a mean rate of 1.24°C d$^{-1}$, or 0.30°C per degree decrease in $\overline{T_a}$. Warming $\overline{T_w}$ in 2023 led to a slight net energy gain in the water column (10.4 - 38.4 W m$^{-2}$) while rapid losses were observed in $\overline{T_w}$ and $T_s$ in 2022. Difference in $\dot{E}$ between 2022 and 2023 peaked on October 12 with a difference of 184.4 W m$^{-2}$. A maximum $\Delta\overline{T_w}$=5.25°C between 2022 and 2023 was observed on October 16$^{th}$ (Fig. 6b). The significantly earlier cooling of the lake in 2022 was attributed to cooler $\overline{T_a}$, and resulted in the 9-day earlier IO as compared to 2023.

$\overline{T_w}$ remained elevated in 2023 until October 20. $\overline{T_w}$ began to decline at a rate of 1.45°C per day from 6.00°C on October 20 to 0.20°C on October 24 with surface temperatures falling below zero on October 21, 2023, leading to the first appearance of ice. Note that $\overline{T_w}$ was not available in 2021.

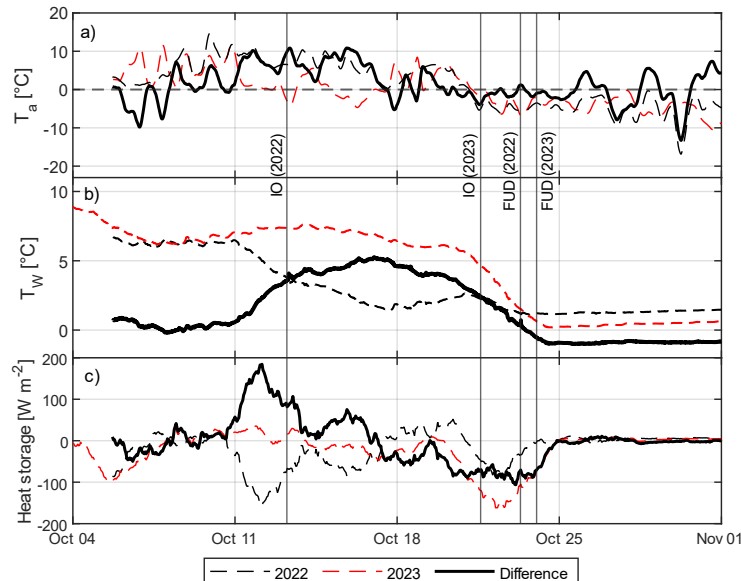

**Figure 6: Variability in a) 15-minute air temperatures ($T_a$), and b) mean water temperatures ($\overline{T_w}$) and c) mean hourly heat storage in 2022 and 2023 recorded at the FRS. Values in a) and c) are smoothed over a 1-day period. The solid black line represents the difference in water or air temperatures between 2022 and 2023.**

In all years, the formation and progression of the ice cover during freeze-up occurred in a similar fashion. Ice first appeared as border ice in the shallower southern sections of Landing Lake and along the shoreline. Mean lake depths in these southern arms were typically less than 1.5 m deep. The ice front processed inwards from the shoreline and northward towards the deeper, central body of the lake until the entire lake was ice covered. Weather variability between the years played a dominant role in controlling freeze-up durations, particularly in 2022 where freeze-up

lasted 11 days (Table 4).

## 5.2. Evolution of ice and snow

Ice and snow evolution in 2021, 2022, and 2023 were distinct. In 2021, ice growth following freeze-up was extremely fast, ~6-8 cm per week as a result of low snowfall (Table 1, Rafat et al., 2023). Ice thickness reached 10 cm on November 15, ~6-8 days after the FUD (Table 5). Ice growth remained fast as dry conditions persisted in

November 2021 with only 14.9 cm of cumulative snowfall having been recorded by the end of November 2021 and manually measured snow depths <10 cm. 30.9 cm of total snowfall was recorded by December 31, 2021. Ice thickness reached 30 cm 31-33 days after the estimated FUD, and 52 cm by January 1, 2022.



**Table 5 : Comparison of ice evolution 2021-2023**

| | Date | | | Duration (days) | | $h_i$ (cm) | | |
|---|---|---|---|---|---|---|---|---|
| | Freeze-up Date | $h_i$=10 cm | $h_i$=30 cm | Freeze-up Date to $h_i$=10 cm | Freeze-up Date to $h_i$=30 cm | Nov. 1 | Dec. 1 | Jan. 1 |
| **2021** | Nov. 7-9[*] | Nov.15[*] | Dec. 2[*] | 6-8 | 31-33 | X | 28 | 52 |
| **2022** | Oct. 23 | Nov. 3 | Nov. 26 | 11 | 34 | 8.2 | 33 | 52 |
| **2023** | Oct. 24 | Nov. 2 | Dec. 4 | 9 | 41 | 9.9 | 27 | 40 |

[*]*Interpolated values pre-deployment of SIMBA on December 6th, 2021*

IO and FUDs in 2022 were significantly earlier (~2 weeks) than in 2021; however, freeze-up durations were similar (11 vs 7-9 days). Earlier IO in 2022 did not result in thicker ice when compared to recorded ice thicknesses in December 2021 due to high snowfall and deeper snow on Landing Lake (Fig. 7a). Interestingly, the duration between the FUD to $h_i$=30 cm in 2022 was nearly identical to 2021 as were the ice thicknesses recorded on December 1 and January 1 (Table 5). Cumulative snowfall in 2022 surpassed 30.9 cm (cumulative snowfall up to

December 31, 2021) on October 29, 2022, only 6 days following the FUD. The high quantity of snowfall and warmer air temperatures in 2022 (compared to 2021) following the FUD likely contributed to the increased duration from the FUD to $h_i$=10 cm.

Freeze-up occurred quickly in 2023 taking 3 days from IO to the FUD. Cumulative snowfall values were greater than in 2021, but less than in 2022. However, snow depths recorded on Landing Lake were significantly higher than

in 2022 (Fig. 7a). This may be explained by frequent slushing events that were recorded following freeze-up in 2022 resulting in relatively shallow snow depths or from snow redistribution effects. Ice thicknesses were lower in 2023 than in both 2022 and 2021. $CFDD$ was greater than 2021 in October and most of November but lower than 2022. In December, warm air temperatures resulted in a significant reduction in $CFDD$ by the end of the month (Table 1). Low $CFDD$ and moderately high snowfall resulted in low ice thicknesses and slow ice growth, taking 41 days for ice

thicknesses to reach 30 cm from the FUD and only growing 13 cm in December.

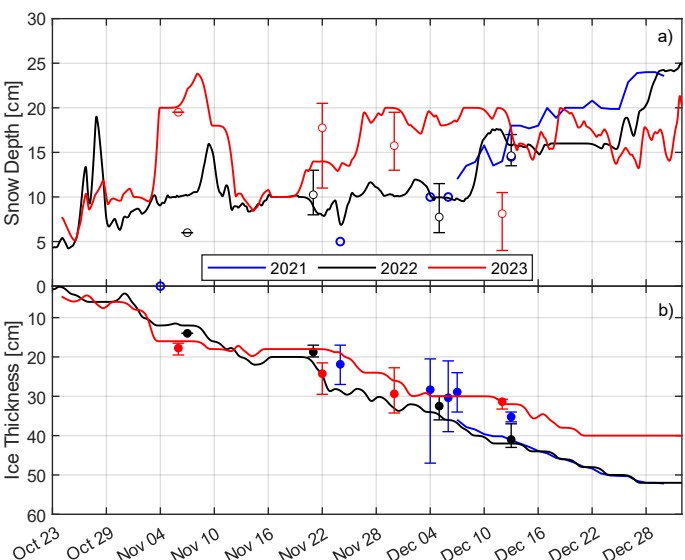

**Figure 7. SIMBA-derived a) snow and b) ice evolution in Landing Lake in 2021, 2022, and 2023 from the freeze-up date through to December 31. Open and closed circles represent the mean value of manual measurements of snow and ice**
**respectively. Error bars show measured spatial variability around the SIMBA. Ice and snow measurements are smoothed over a 3-day sliding window.**

## 6. Relating air, snow, and ice evolution

Variability in air temperatures and snow resulted in three unique responses in the timings of IOs, FUDs, and the growth of ice. 2021 was classified as a high $CFDD$, low snowfall year. 2022 showed near-normal (± 15% of 1982-
2010 climate) $CFDD$ but showed above-average (>15%) total snowfall $S_T$. 2023 presented the case where end of December $S_T$ was only 106% of normal, but end of December cumulative $FDD$ was only 74% of normal. The effects of air temperature on ice growth are commonly represented using Stefan's Equation (Eq. 5a). $CFDD$ was observed to have an exponential relationship with $S_T$ (Fig. 8a) in the form of Eq. (5b). Hence, ice thicknesses may be explicitly modelled as an exponential function of snowfall (Eq. 5c), and indirectly as a function of time. Equation
6c may be further simplified into a two-constant model by setting $C = \alpha \left( \sqrt{\frac{2k_i}{\rho_i L}} \right) a$, such that Equation 5c becomes $h_i = Ce^{0.5bS_T}$ (Eq. 6). $\alpha$ was determined by minimizing the root mean square error (RMSE) of modelled and measured ice thicknesses in 2021, 2022, and 2023.

$$h_i = \alpha \sqrt{\frac{2k_i}{\rho_i L} \cdot CFDD} \tag{5a}$$

$$FDD = ae^{bS_T} \tag{5b}$$





$$h_i = \alpha\left(\sqrt{\frac{2k_i}{\rho_i L}}\right)ae^{0.5bS_T} \tag{5c}$$

$$h_i = Ce^{0.5bS_T} \tag{6}$$

Initial RMSE between modelled and measured ice thicknesses were small (< 3.1 cm), but deviations in FUDs were
large (7-29 days). Deviations were large due to varying latency effects associated with lake cooling. Equations 5c
and 6 do not explicitly include a melt or retardation factor to account for air temperatures >0°C and consider IO
equivalent to FUDs. Hence, sub-zero air temperatures which occur well before observed FUDs (as in 2021 and
2022) result in modelled FUDs to be well in advance to observed FUDs. Moreover, this approach does not consider
latency between changes in lake surface water temperatures and air temperatures. Air temperatures may fall below
0°C triggering Equation 5c and 6 to produce ice yet, Landing Lake still may continue ample heat to prevent ice
formation. To account for these effects within Landing Lake the baseline temperature (BT) for which $CFDD$ is
calculated was reduced from 0°C to -5°C. The finalized empirical model using BT= -5°C is presented in Fig. 8.

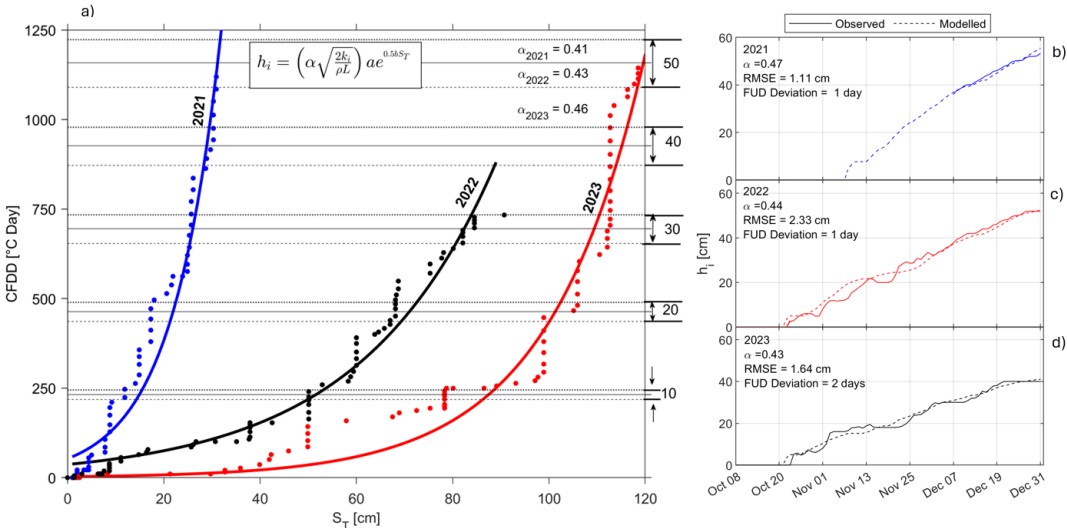

**Figure 8: Relationships between cumulative freezing degree days ($CFDD$), cumulative snowfall ($S_T$) and ice thicknesses
($h_i$) for 2021 (blue), 2022 (red), and 2023 (black). Dashed, solid, and dotted horizontal lines in a) represent values of α in
years 2021, 2022, and 2023 respectively. Dashed and solid lines in b), c) and d) represent modelled and measured ice
thicknesses respectively.**

BT=-5°C was selected as it provided the lowest error (RMSE and deviation) in modelled versus observed $h_i$ and
FUDs in the range of -10°C ≤ BT < 0°C (Fig. 9). Optimized values of $\alpha$, $a$, $b$, and $C$ for years 2021, 2022, and
2023 are summarized in Table 6. Variations in the value of all constant under varying BTs are presented in Table
A1. While BT=0°C is commonly used (*e.gs:* Gow & Govoni, 1983; Michel, 1971), the choice of 0°C as a threshold
for calculation of $CFDD$ in lakes is arbitrary, with any sub-freezing temperature proving sufficient. Interestingly,
here, we note that BT=-5°C provided the lowest RMSE and deviation in $h_i$ and FUD across all years. This baseline





is colder than that used commonly used for sea ice of BT= -1.8°C for salinity of 32‰ (Bilello, 1961, ISO, 2019).
Although only considering ice melt in his analysis, Bilello (1980) provides a through discussion on the use of 0°C, -
1.8°C, -5°C and -10°C as BTs for evaluating cumulative simulations of ice decay using thawing-degree days
(CTDD). Bilello (1980) concluded that the use of BT=0°C was most appropriate for simulation of break-up using
CTDD in lakes, and -5°C in rivers citing melt occurring before air temperatures rise to 0°C. The inverse argument
can be applied to the freezing process where lakes and rivers do not necessarily freeze immediately following air
temperatures falling below 0°C. It is coincidental that our findings presented the lowest error for CFDD calculating
using BT=-5°C, the optimal threshold for CTDD in rivers identified by Bilello (1980).

**Table 6: Constants for $CFDD$ model using BT = -5°C**

|  | Year | | |
|---|---|---|---|
| **Constant** | **2021** | **2022** | **2023** |
| $\alpha$ | 0.47 | 0.44 | 0.43 |
| $a$ | 52.87 | 2.91 | 36.86 |
| $b$ | 0.099 | 0.050 | 0.036 |
| $C$ | 0.903 | 0.047 | 0.576 |

$\alpha$ and $b$ from the final model decreased with decreasing end of December $CFDD$ and generally decreased with
increasing $S_T$. This agreed with the understanding that $\alpha$ decreases with increasing snow and flow (Michel, 1971;
Shen, 2010). The sensitivity of $\alpha$ and $b$ to $S_T$ is not linear however, as $\alpha$ and $b$ in 2022 were slightly larger than in
2023 despite end of December $S_T$ in 2022 being larger than in 2023. $\alpha$ and $b$ were observed to decrease consistently
with increasing $h_s$ (Fig. 7) and snow-ice ($h_{si}$) (Sect. 5.2). Both relationships are logical as more snowfall generally
increases snow depths over lakes (albeit with significant variability) which increases the likelihood of snow-ice
formation.

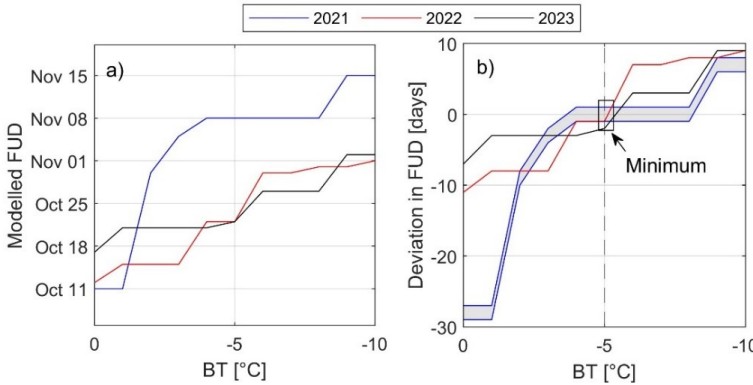

**Figure 9: Effect of shifting baseline temperature (BT) on a) modelled FUDs, and b) deviations between modelled and**
**observed FUDs. The grey shaded region represents the uncertainty in FUD sensitivity to BT from uncertainties in the**
**estimated FUD in 2021.**

$\alpha$ showed no sensitivity for decreasing $CFDD$ BT from 0°C to -5°C but linearly increased from -5°C to -10°C. The
greatest sensitivity was observed in 2023 in the range -5°C to -10°C with $\alpha$ increasing by an average of 4.9% per °C




reduction in BT. Sensitivity in 2021 and 2022 was marginal but slightly larger than in 2022. $a$ showed a strong
decreasing trend with decreasing baseline temperatures across all years. A similar trend was observed for $a$ in 2023
which had the highest sensitivity to declining baseline temperatures with an increase of 11.6% per °C between -10°C
to -5°C. $b$ showed slight increasing trends with decreasing baseline temperatures over the entire range of tested BT.
The magnitude of $C$ decreased with decreasing BT across all years (Table A1).

Snow depths ($h_s$) over lake ice were linearly related to $S_T$ in all years (Fig. 10). Correlations were strongest in 2021
($r^2$=0.82) and 2023 ($r^2$= 0.75), and the weakest in 2022 ($r^2$=0.59). The strong correlation in 2021 was likely
attributed to no snow-ice being produced and generally low $S_T$. Both 2022 and 2023 had mean proportions of snow-
ice to total ice $\left(\frac{h_{si}}{h_i}\right)$ of 18% (0-54%) and 33% (0-44%). Interestingly, the correlation between $h_s$ and $S_T$ was
stronger in 2023 than in 2022, despite $\frac{h_{si}}{h_i}$ being greater in 2023. This finding suggests that snow-ice formation does
not account for all observed variability. The remaining variability may be attributed to snow redistribution and
metamorphic processes which can create significant spatial and temporal variability for on lake snow-depths (Pouw
et al., 2023). Moderately positive correlations ($r^2$=0.50, 0.67, and 0.76 for 2021, 2022, and 2023) were also observed
between $h_s$ and $h_i$. This finding was as partially unexpected as deeper snow over ice slows ice growth provided
snow-ice is not formed and density remains relatively constant. The positive correlation may thus allude to the
positive contribution of snow to snow-ice formation.

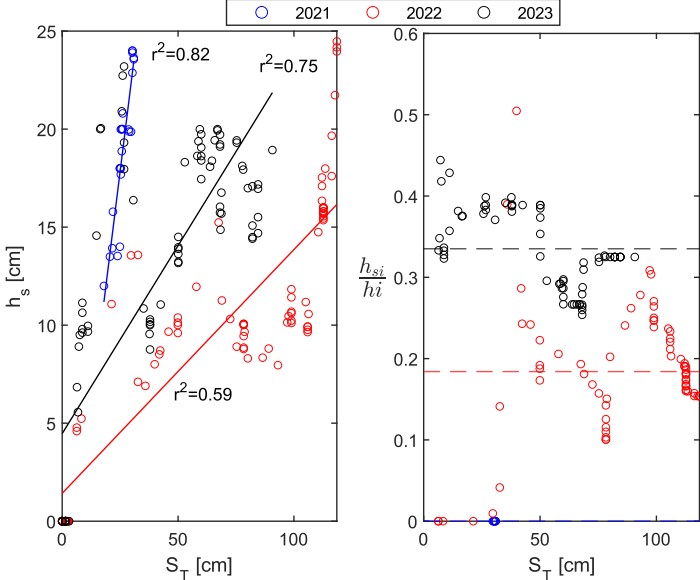

**Figure 10: Correlations between a) snow depths, and b) proportions of snow-ice to total ice against total snowfall.**

The ability to accurately simulate snow-ice thickness is useful for understanding the load bearing capacity of ice
covers. It was observed that $h_{si}$ in both 2022 and 2023 could be effectively reproduced using a simple multi-linear
regression model in the form of $h_{si} = D_1 + D_2 S_{day} + D_3 h_i$. The model consisted of 3 constants ($D_1$, $D_2$ and $D_3$) and






only 2 variables, daily snowfall (non-cumulative, $S_{day}$) and $h_i$. (Fig. 11). Using the model, RMSEs in 2022 and
2023 compared against measured $h_{si}$ were 1.21 and 0.81 cm respectively. The accuracy of the simulations was
further evaluated using the Nash-Sutcliffe efficiency (NSE) parameter, with both years showing excellent simulation
strength (NSE=0.86 and 0.95, 2022 and 2023 respectively). Values of constants $D_1, D_2$, and $D_3$ are presented in
Table 7.

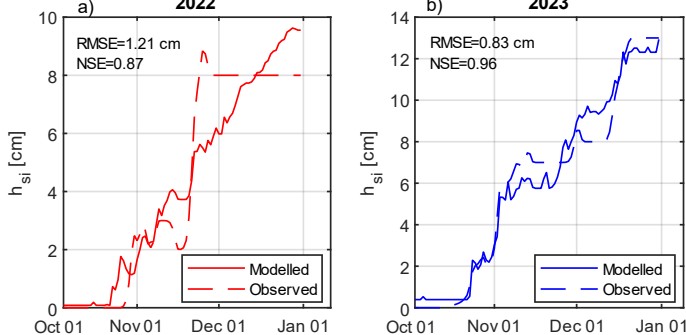

**Figure 11: Modelled versus observed snow-ice thicknesses**

**Table 7: Value of constants for the linear snow-ice model**

| Constant | Year | |
| --- | --- | --- |
|  | 2022 | 2023 |
| $D_1$ | 0.0826 | 0.3917 |
| $D_2$ | 0.0521 | 0.0100 |
| $D_3$ | 0.1821 | 0.2980 |

$h_{si}$ was more accurately simulated in 2023 as compared to 2022. This finding is likely the result of the available ice
freeboard during the time of the daily snowfall. In 2023, higher snowfall was closely followed by snow-ice
formation (Fig.12b) suggesting near-critical submergence conditions (low freeboard). This contrasts with 2022
where between November 4-18, 23.1 cm of snowfall was recorded but no snow-ice was produced (Fig. 12a)
suggesting that ample freeboard was present in the ice cover to support significant snow loading without
submergence. This phenomenon is reflected in Fig. 12a where modelled $h_{si}$ was not able to accurately capture the
rapid snow-ice formation between November 19-25 when the available freeboard was thought to be exceeded. The
addition of a freeboard component to the model would likely improve simulation strength but at a cost of increased
complexity and uncertainty as the accurate estimation of freeboard is a non-trivial task.



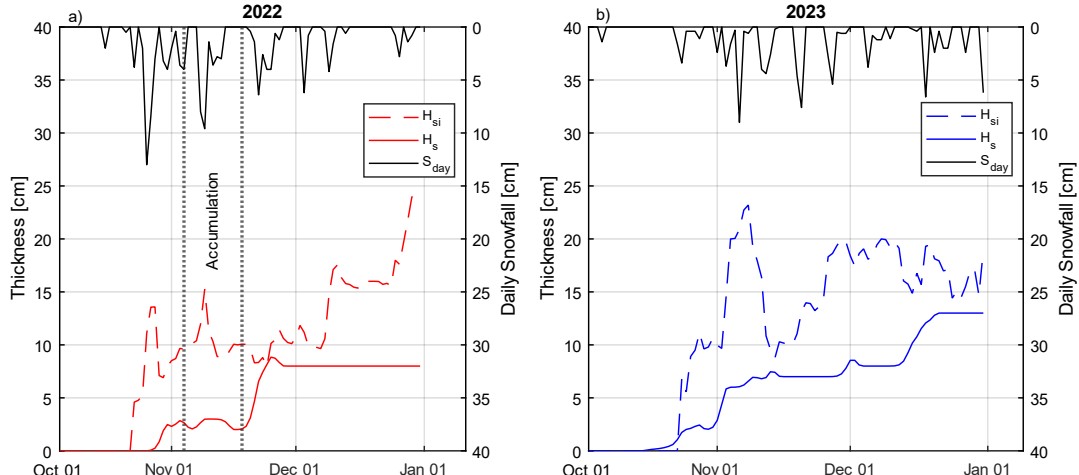

**Figure 12: Interactions between snowfall, snow depths, and snow-ice in a) 2022 and b) 2023.**

## 7. Conclusions

In this study we investigated the influence of weather on the evolution of ice in a small and shallow subarctic lake between September-December 2021, 2022, and 2023. Weather variability was described using air temperature and snowfall. Combinations of varying air temperatures and snowfall conditions resulted in three unique responses in the timings of IOs, FUDs, and the growth of ice for 2021, 2022, and 2023 respectively. Variability of up to 20 days in IOs, 17 days in FUDs, and 8 days in freeze-up durations were observed. The duration between FUDs and when ice thicknesses ($h_i$) reached 30 cm varied between 31-41 days, while the timing from FUDs to $h_i$ =10 cm varied between 6-11 days. Ice thicknesses on December 1 varied by only 6 cm between the years (27-33 cm) but doubled by December 31 to 12 cm (40 - 52 cm). Changes in water temperatures closely followed changes in air temperatures which controlled the timing of FUDs, yet the crossing of the zero-degree isotherm was observed as not being a reliable indicator for use in predicting IOs or FUDs.

Variability in ice evolutions between 2021, 2022, and 2023 were effectively explained using an empirically derived model involving cumulative freezing degree days ($CFDD$) and snowfall ($S_T$) in the form of $h_i = \alpha \left( \sqrt{\frac{2k_i}{\rho_i L}} \right) a e^{0.5bS_T} = C e^{0.5bS_T}$, where $\alpha$, $a$, and $C$ are constants. $h_i$ were effectively simulated in all years with RMSE < 2.33 cm, with accuracy in estimated FUDs of $\leq$ 2 days when calculating $CFDD$ using a -5°C threshold. A simple model for simulation of snow-ice thicknesses using $CFDD$ and daily snowfall ($S_{day}$) in the form of $h_{si} = D_1 + D_2 S_{day} + D_3 h_i$ proved effective (RMSE $\leq$ 1.21cm), where $D_1, D_2$ and $D_3$ are fitted constants. Snow depths over lake ice were found to be linearly related to $S_T$ (r²=0.59-0.82) with the strength of the correlations decreasing with increasing $S_T$. Developed empirical relationships may be site-specific, but are simple, and useful means of anticipating ice growth given short term forecasts of snowfall conditions and air temperatures. With scattered measurements of ice thicknesses in a particular year, the derived relationships can be used by practitioners as first order estimates for simulating ice thicknesses under weather variability.



## 8. Applications and significance

Under future climate change, winter precipitation in northern Canada is projected to increase (Zhang et al., 2019).
Predictions of changes in snowfall can be used with the presented empirical relationships to understand, to a first-order, potential shifts in ice thicknesses and composition with climate change. Estimates of snow-ice thicknesses and proportions of snow-ice to total ice thickness are significant for: 1) estimating future bearing capacity of ice covers and for adaptation of ice road designs, 2) predicting future BUDs, and 3) understanding possible changes to under-ice ecological processes.

Following 1), Gold's formula remains the standard approach for bearing capacity estimates for ice road design, with a reduction to the effective ice thickness to compensate for snow-ice of lower quality (less dense; Masterson, 2009). Hence, estimates of snow-ice proportions may be used (with caution), as a first-order approximation of future changes in load bearing capacities. However, this approach has limitations. While the strength of snow-ice under confinement may be lower than congelation ice, the strength of an ice sheet undergoing bending or punching failure,
composing of varying proportions of snow and congelation ice is not-well investigated. Additionally, the effects of varying ice layers on ice flexural strength are thought to be variable (Daly et al., 2023). Note that snow-ice may be considered equivalent in strength to congelation ice for densities >880 kg m$^{-3}$ (Masterson, 2009) highlighting the nuances of snow-ice in bearing capacity estimation. There is hence a growing need to re-visit and modernize Gold's formula in response to current climate change, perhaps through reproducing large-scale breakthrough testing
conducted in Canada prior to the 1960s (Gold, 1960) taking into account ice-composition. Future efforts to modernize routine ice road operations may choose to recognize said limitations in Gold's formula (*e.g.* Fitzgerald and van Rensburg, 2024) and adopt a limit stress based approach (Masterson, 2009).

Greater snow-ice thicknesses may delay BUDs in lakes as snow-ice effectively scatters insolation and may slow internal melting and deterioration. Rapid deterioration of ice in the spring leads to increasing porosity and rapid
collapse of the ice sheet (collapse failure) as was evident during the decay process in Landing Lake in May 2023 (Rafat et al., 2024). Greater snow-ice thicknesses may also prevent this failure mechanism through reducing the candling of the ice cover. This effect may be exploited for late season ice road operations. For instance, one lane of road traffic can be closed and covered with compacted snow or flooded to form to snow-ice while the other lane is used. Once the active lane is degraded, traffic can be re-directed to the snow-ice/snow covered lane (upon clearing)
(Strandberg et al., 2012). Said lane will have its strength properties largely intact due to reduced internal melt and deterioration. For snow-ice thicknesses greater than 30 cm the magnitude of photosynthetically active radiation that is able to penetrate the ice cover is approaches zero (Kirillin et al., 2012). This directly influences under-ice aquatic ecology through influencing lake mixing and primary productivity (Hampton et al., 2017).

Understanding variability in early winter ice formation and growth is essential for ice road design. Empirical
relationships between $CFDD$ and $S_T$ may allow engineers to select approach construction methods and equipment, establish appropriate quality and hazard control plans, and determine if critical conditions may exist to warrant expanded stress-state analyses or interventions during ice road constuction. Models presented in Fig. 9 and Fig. 11 can be used as templates for understanding , as a first-order approximation, ice evolution using early winter weather and site specific characteristics.



**Code and data availability**

Data used to generate conclusions in this study can be found will be made available in an appropriate repository. Air temperatures and snowfall measurements between 1942-2023 at the Yellowknife Airport weather station can be readily accessible from Environment and Climate Change Canada (https://climate.weather.gc.ca/). Code used for conducting analyses in this study are available from the corresponding author upon request.

**Author Contribution**

AR: data collection, data processing, writing-original draft, HKP: Supervision, resources, Writing - Review & Editing.

**Competing interests:**

One of the authors is a member of the editorial board of *The Cryosphere*.

**Acknowledgements:**

This research was supported by the Government of Northwest Territories, Environment and Climate Change, Cumulative Impact Monitoring Program (CIMP-212), Natural Sciences and Engineering Research Council of Canada (NSERC) Canada Research Chair (CRC) and Discovery Grant (RGPIN-2020-05573), the Polar Knowledge Canada Northern Scientific Training Program (NSTP), and the NSERC Vanier Graduate Scholarship.

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



## Appendix A

**Table A1: Sensitivity of modelled FUD, $\alpha$, and RMSE to $CFDD$ baseline temperature (BT) (°C)**

| BT (°C) | FUD | | | RMSE | | | $\alpha$ | | | $a$ | | | $b$ (x10²) | | | $c$ | | |
|---|---|---|---|---|---|---|---|---|---|---|---|---|---|---|---|---|---|---|
| | 2021 | 2022 | 2023 | 2021 | 2022 | 2023 | 2021 | 2022 | 2023 | 2021 | 2022 | 2023 | 2021 | 2022 | 2023 | 2021 | 2022 | 2023 |
| 0 | Oct 11 | Oct 12 | Oct 17 | 1.08 | 3.04 | 1.95 | 0.46 | 0.43 | 0.41 | 59.20 | 4.66 | 45.00 | 9.60 | 4.60 | 3.40 | 0.990 | 0.073 | 0.671 |
| -1 | Oct 11 | Oct 15 | Oct 21 | 1.09 | 3.01 | 1.93 | 0.46 | 0.43 | 0.41 | 59.10 | 4.62 | 44.93 | 9.60 | 4.60 | 3.40 | 0.997 | 0.072 | 0.670 |
| -2 | Oct 30 | Oct 15 | Oct 21 | 1.10 | 2.98 | 1.93 | 0.47 | 0.43 | 0.41 | 57.02 | 4.50 | 44.93 | 9.70 | 4.65 | 3.40 | 0.974 | 0.070 | 0.670 |
| -3 | Nov 05 | Oct 15 | Oct 21 | 1.07 | 2.72 | 1.93 | 0.47 | 0.43 | 0.41 | 55.32 | 3.09 | 44.54 | 9.79 | 4.77 | 3.40 | 0.945 | 0.048 | 0.664 |
| -4 | Nov 08 | Oct 22 | Oct 21 | 1.11 | 2.41 | 1.88 | 0.47 | 0.44 | 0.41 | 52.87 | 3.35 | 42.96 | 9.90 | 4.89 | 3.43 | 0.903 | 0.054 | 0.640 |
| -5 | Nov 08 | Oct 22 | Oct 22 | 1.11 | 2.33 | 1.64 | 0.47 | 0.44 | 0.43 | 52.87 | 2.91 | 36.86 | 9.90 | 5.01 | 3.57 | 0.903 | 0.047 | 0.576 |
| -6 | Nov 08 | Oct 30 | Oct 27 | 1.11 | 2.43 | 1.95 | 0.47 | 0.46 | 0.44 | 52.87 | 1.65 | 33.00 | 9.90 | 5.47 | 3.67 | 0.903 | 0.028 | 0.528 |
| -7 | Nov 08 | Oct 30 | Oct 27 | 1.17 | 2.61 | 2.05 | 0.47 | 0.47 | 0.47 | 51.47 | 1.00 | 27.01 | 9.97 | 5.88 | 3.80 | 0.879 | 0.017 | 0.462 |
| -8 | Nov 08 | Oct 31 | Oct 27 | 1.23 | 2.89 | 3.35 | 0.48 | 0.48 | 0.49 | 46.93 | 0.70 | 19.91 | 10.2 | 6.00 | 4.09 | 0.819 | 0.012 | 0.355 |
| -9 | Nov 15 | Oct 31 | Nov 02 | 1.38 | 2.89 | 5.20 | 0.49 | 0.48 | 0.51 | 43.48 | 0.70 | 15.93 | 10.4 | 6.16 | 4.29 | 0.775 | 0.012 | 0.295 |
| -10 | Nov 15 | Nov 01 | Nov 02 | 1.38 | 3.13 | 5.12 | 0.49 | 0.49 | 0.53 | 43.48 | 0.41 | 14.25 | 10.4 | 6.59 | 4.36 | 0.775 | 0.007 | 0.275 |
