# Peer review of "Interannual variability in air temperature and snow drive differences in ice formation and growth"

_EGUsphere, 2025_

## Author Comment (AC1)

Response to Reviewer 1:

**Reviewer comments are provided in black.**
**Author responses are provided in blue**

The manuscript presents an empirical model for lake ice formation and growth based on three-year field observations at Landing Lake, Canada. While the methodology demonstrates potential for winter road management and climate change monitoring, several critical issues require clarification to strengthen scientific rigor and practical applicability. Specific recommendations are organized as follows:

**Specific comments:**

1. Line 61: Correct "Xinjing" to "Xinjiang"

   This spelling error has been corrected.

2. Lines 39-71: Condense discussions on ice phenology studies.

   We thank the Reviewer for this suggestion. We have correspondingly condensed the discussion on ice phenology as follows:

   *Phenological changes in lake ice covers have been explored across many northern, high-latitude regions and strongly relate to weather conditions (Huang et al., 2019; Latifovic and Pouliot, 2007; Leppäranta et al., 2017). There is coherence amongst most published literature that lakes across the northern hemisphere are experiencing earlier break-up dates (BUDs), with some exceptions depending on time periods analysed, significance levels attributed to trends, and specific regions. Trends of earlier BUDs have been observed in Canada between 1961-1990 (Duguay et al.,2006) in Sweden between 1870-2010 (Hallerbäck et al., 2022; L'Abée-Lund et al., 2021), Poland (1961-2010; Choiński et al., 2015), Lake Baikal (1869-1996; Todd & Mackay, 2003), and in the Laurentian Great Lakes Region (1975–2004, Jensen et al. 2007). Meta-analysed conducted by Newton and Mullan (2021) and studies derived from the Global Lake and River Ice Phenology (GLRIP) Dataset produced by Benson et al. (2002) spread mostly across North America and northern Europe show similar results.*

   *Trends in freeze-up dates (FUDs) have shown much greater spatial variability, as ice formation depends strongly on local topography, lake morphology, and lake heat storage (Leppäranta, 2015).Regional trends in FUDs are often masked out, under-represented, or are not available, particularily in meta-analyses where a majority of lake may show later FUDs (e.gs. Sharma et al., 2021; Basu et al., 2024). Within meta-analyses, definitions used for delineating FUDs, and methods of observation of ice formation vary in space and with time creating a challenge for drawing accruate conclusions (Catchpole and Moodie, 1974; Wynne, 2000), as does the length of available data record (Benson et al., 2012; Supplementary Material; Sharma et al., 2021; Supplementary Material). Notable examples of lakes with earlier observed trends in FUDs in Finnish Lapland (1930s-1960s; Korhonen, 2006), Xinjiang (2001-2018; Cai et al., 2020), eastern Canada (1961-1990) and the Great Lakes-St. Lawrence regions (1951-1980) (Duguay et al., 2006), Kazakistan and Tajikistan (2002-2022; Hou et al., 2022), Lativa (1945-2002; Apsīte et al. 2014), Poland (1960-1989; Girjatowicz et al., 2022), Sweden (1913-2014; Hallerbäck et al., 2022), and in the Qinghai-Tibetian Plateau (2002-2021; Sun et al., 2023; 2000-2011; Yao et al., 2016). Trends of earlier FUDs in the last 30 years are of particular interest as they largely contrast findings presented in Newton and Mullan (2021) and Sharma et al. (2021) who argue synchronicity in later freeze-up dates.*

Response to Reviewer 1:

3. Lines 72-77: Expand on the disadvantages of conventional techniques (manual observations and numerical modeling) compared to the FRS system to emphasize research significance. (1) Manual measurements: Labor-intensive with discontinuous temporal coverage. (2) Numerical models: Computationally demanding.

   This is a welcomed suggestion. We have added the following text to describe the disadvantages of conventional techniques while expanding on how the FRS addresses these disadvantages.

   *Conventionally, high frequency manual measurements of ice thicknesses and snow depths are constrained by finances, labour, site-access, and ice safety and often result in discontinuous datasets. Numerical modelling provides a continuous alternative to frequent in-situ measurements; however, models may be computationally constrained and still require frequent in-situ observations for appropriate calibration. The use of a Floating Research Station addresses these limitations through offering a cost-effective method to measure ice thicknesses and snow depths at high frequencies, without safety constraints, and can provide the necessary in-situ data to calibrate numerical models.*

4. The study focuses on freeze-up, ice-onset, and ice growth. However, the capability of the FRS system to monitor the complete ice thickness cycle (including break-up and melt processes) remains unclear.

   We thank the Reviewer for this inquiry. The Floating Research Station (FRS) was first installed in October 2022 and has since successfully monitored ice thicknesses evolution throughout the year, including during the melt and break-up periods. While this study focuses especially on freeze-up, ice on-set, and ice growth, we acknowledge that a comprehensive analysis for melt processes, and break-up is outside the scope of the present study. However, we have demonstrated the capability of the FRS during the melt and break-up periods in Rafat et al. (2024), where a detailed examination of the FRS during melt and break-up and the recorded ice thicknesses is provided.

   *Rafat, A., Kheyrollah Pour, H., Spence, C., and Palmer, M. J. 2024. A field study of lake ice decay. Proceedings of the 27th IAHR Symposium on Ice. Gdańsk, Poland.*

5. While the manuscript emphasizes the importance of ice simulation for winter road management under climate change, the empirical model is derived from a small lake (1.07 km²). Address whether such site-specific relationships can be generalized to larger water bodies or regions with distinct climatic/hydrological conditions.

   This comment is appreciated. To address limitations and applicability of the empirical models, we have added in a small paragraph in the newly created discussion section of the manuscript. This section of text is provided below for your reference.

   *The presented empirical models demonstrate an effective means of simulating total ice and snow ice thicknesses in Landing Lake using snowfall and air temperatures recorded from the Yellowknife Airport weather station, located 11 km south of Landing Lake. Relationships between CFDD and $S_T$ (Fig. 8) can be considered as regional relationships which can be applied to other Yellowknife-area small lakes with similar lake depths (e.g. < 5 m) and surface areas (e.g. < 5 km²) for first order estimates of ice thicknesses. Further, the same methodology can be applied for establishing values of $\alpha$ $\gamma$, b, C, $D_1$, $D_2$, and $D_3$ in other regions of the Northwest Territories if measurements of snowfall, air temperatures, and a few measurements of ice thickness are available. This analysis is not intended for use in large and deep lakes whose latency effects during freeze-up would require*

Response to Reviewer 1:

> *unique treatment. Multi-year monitoring in other regions of the Northwest Territories can aid in establishing regional curves such as those presented in Fig. 8 for determining inter-annual and regional variability in model parameters.*

We note further that to demonstrate the applicability of the model, we have applied the model to Vee Lake, a lake with similar geometric properties to Landing Lake. Further details are provided in our response to Reviewer Comment 23.

6. Lines 96-101: Add references to support statements about meteorological data requirements.

We have added references supporting the description of the meteorological data in Yellowknife, Northwest Territories, Canada,

*Spence, C. and Hedstrom, N.: Hydrometeorological data from Baker Creek Research Watershed, Northwest Territories, Canada, Earth Syst. Sci. Data, 10, 1753–1767, https://doi.org/10.5194/essd-10-1753-2018, 2018.*

*Environment Canada. Climate Data Online. https://climate.weather.gc.ca/climate_normals/index_e.html, 2025. [April 4, 2025].*

7. Section 3.1.1: Clarify MODIS data usage: (1) Specify product version (MOD11A1/MYD11A1?). (2) Justify spatial representativeness: How were pixel quality issues addressed for a 1.07 km² lake under 1 km resolution? (3) Indicate whether day or night data were used.

1) The MODIS/Aqua Land Surface Temperature/Emissivity Daily L3 Global 1-km SIN Grid product (MYD11A1) was used for reference in this study. 2) We acknowledge the pixel quality challenges associated with a 1.07 km$^2$ lake and a 1 km$^2$ grid and potential land inclusions within the pixel. Despite these limitations, the MYD11A1 product was still the optimal choice given the frequency of measurements. As per your suggestion, the MODIS data was compared to optical Sentinel-2 imagery and found good agreement between <0°C surface temperatures and ice appearance on the lake justifying the appropriateness of the selection. 3) Only day-time values were used.

8. Line 167: State the distance between Yellowknife Airport station and Landing Lake.

The distance (11 km) between the Yellowknife Airport weather station and Landing Lake has been added to this line.

9. Lines 278-288: While agreeing with the 2021 ice-onset (IO) and freeze-up date (FUD) determinations, we recommend utilizing Sentinel-2 imagery for independent validation.

This is a helpful suggestion. We had previously independently verified the IO and FUD using Sentinel-2 optical imagery which agreed well with these determinations. We had made note of this on L155/156 of the submitted manuscript.

10. Figure 5a: (1) Explain discontinuous $T_s$ curves: Were data gaps caused by cloud masking or quality filtering? (2) Replace connected lines with discrete markers (e.g., circles) for non-continuous MODIS data.

Response to Reviewer 1:

The discontinuous $T_s$ curve in Figure 5a was the result of both cloud masking and quality filtering of the MODIS/Aqua Land Surface Temperature/Emissivity Daily L3 Global 1-km SIN Grid product (MYD11A1). As suggested, we have replaced the previously connected curve in Figure 5a with only the discrete, MODIS-derived $T_s$.

11. Definition inconsistency: Ice-onset (IO) and freeze-up dates (FUD) are defined as horizontal lake-wide phenomena (Line 18), yet 2022–2023 determinations rely on vertical SIMBA temperature profiles.

We appreciate the comment and here we are providing further clarity. In 2021, ice-onset and FUDs were defined according to MYD11A1 surface temperatures and Sentinel 2 optical imagery since SIMBA measurements were not available during freeze-up. This had resulted in IO and FUD being defined horizontally as noted by the Reviewer's comment. In 2022 and 2023, SIMBA was used to determine when the surface, at the location of the FRS, had frozen. The IO and FUD were not defined vertically by a threshold ice thickness (e.g. > 3cm), but rather by the appearance of ice on the surface measured by the SIMBA. To rectify differences between 2021 and 2022/2023, we note that the FRS was deployed within the deepest part of Landing Lake and is commonly one of the last locations to freeze. The ice front in Landing Lake progresses from the southern, shallower 'arms' and from the shoreline and continues northward towards the FRS- we describe this progression on L30-35. As such, if ice is identified at the surface near the FRS, it is highly likely that most of the lake is frozen over, thereby additionally satisfying the 'horizontal' criterion implicitly. We hope this satisfies this Reviewer's inquiry.

12. Figure 6: Include time-series plots of SIMBA-recorded vertical thermal profiles to illustrate water column stratification dynamics.

We thank the Reviewer for this suggestion. From investigating select temperature profiles measured by the SIMBA in 2022 and 2023 we noted that: 1) no significant thermal stratification occurs during the open-water season (before FUD) as the lake depth at the FRS is only ~3 m deep and the water well-mixed. Only minor stratification occurs in the upper 6 cm of the lake during the cooling period immediate before IO. 2) Once an ice cover is present, inverse stratification occurs as heat stored in sediments is released over the winter as is typical during the winter periods for shallow lakes. To demonstrate these dynamics, two references figures (Figure R1 and R2) were prepared and provided in response to the Reviewer's comment. While we agree that the SIMBA-recorded temperature profiles may be of interest to some readers, based on the analysis, we have decided not to include an additional subplot for SIMBA time series of thermal profiles within Figure 6.

13. Table 4: Small lakes exhibit low thermal inertia, leading to rapid air temperature responses (11- and 3-day freeze-up durations in 2022–2023). However, the stable water temperature ($T_w$) in 2022 contradicts this pattern. Analyze potential causes: (1) Assess vertical stratification using mixed-layer depth calculations. (2) Evaluate whether the lake remained fully mixed.

We thank the Reviewer for this comment. The Reviewer is correct in stating that small lakes have low thermal inertia, as was observed in this study; however, we would like to note that $T_w$ in 2022 does not completely contradict this pattern. The FUD, IO, and the freeze-up durations described in Table 4 are consequences of surface temperatures ($T_s$) falling below 0°C and not from the mean water temperatures presented in Figure 6. Mean $T_w$ in Figure 6b for 2022 do in fact react quite abruptly to large changes in air temperatures, agreeing with the low thermal inertia statement. On L317 we make note of this change: " $\overline{T_w}$ in 2022 declined beginning October 10 at a mean rate of 1.24°C d$^{-1}$, or 0.30°C per degree decrease in $\overline{T_a}$." $T_w$ was relatively stable, as the Reviewer describes,

Response to Reviewer 1:

prior to October 11 at ~6.3°C. However, small perturbations were present in response to cooling. This was evident during cooling of $T_a$ on October 5-6[th] when $T_a$ decreased from ~3.3°C to 0.65°C and mean $T_w$ decreased from 6.7°C to 6.3°C. (Figure R1). The effects of cooling $T_a$ were more evident for $T_s$ (Figure 5b) than for $T_w$ (Figure 6b). The consequence of this was a ~ 6 cm layer of stratified water at the surface. Figure R1 below shows this phenomenon in 2022 and Figure R2 for 2023. This surface stratified layer is not to be confused with the friction boundary layer immediately above the water surface. We have identified the water surfaces in each of these figures to help delineate between the two.

[Figure]

**Figure R1: Stratification dynamics in response to cooling air temperatures prior to ice-onset in 2022**

[Figure]

**Figure R2: Stratification dynamics in response to cooling air temperatures prior to ice-onset in 2023**

Response to Reviewer 1:

As is evident in Figures R1 and R2, only a weak stratified layer forms on the surface. Deeper waters remain fully fixed. Stable, inverse stratification forms after the FUDs in both years and remains this way for the remaining study period.

14. Figure 7a: Explain the abrupt snow depth reduction on 7 November 2023 (25 cm → < 10 cm). Was this due to melting, compaction, or sensor artifacts?

The reduction in snow depths beginning on November 7[th], 2023, was largely from wind scour and redistribution. A significant amount of fresh snowfall (9 cm) had fallen on November 6[th] corresponding to the large peak in snow depths on November 7[th] over Landing Lake. Although visually the reduction appears quite abrupt, the timescale over this large reduction in snow depths from 22.7 cm to 8.8 cm is 7 days over which significant wind gusts up to 13.6 m s$^{-1}$ (November 9[th], 18:15) were recorded. It is therefore likely that much of the freshly fallen snow was redistributed to elsewhere on the lake, away from the FRS. From Figure 12b, it is also evident that ~1 cm of snow-ice had formed during this period which would corresponding aid to reducing the snow depth. Note that Figure 12 in the original manuscript had a mislabeled legend. The dashed lines represent snow depths and solid lines snow-ice thicknesses. This correction has been made in the revised manuscript.

15. Equations 5a–5b: Replace ambiguous coefficient symbols (e.g., use β, γ instead of α, a) to avoid confusion.

This is a great suggestion and recognize that these coefficients may be confusing, especially $\alpha$ and $a$. We have replaced $a$ to be $\gamma$ but have retained the use of $\alpha$. We chose to retain $\alpha$ as it is commonly used when discussing *CFDD* models and Stefan's Equation.

Equations 5b, c and 6 now read:

$$FDD = \gamma\, e^{bS_T} \tag{5b}$$

$$h_i = \alpha \left( \sqrt{\frac{2k_i}{\rho_i L}} \right) \gamma\, e^{0.5bS_T} \tag{5c}$$

$$h_i = C e^{0.5bS_T} \tag{6}$$

16. Line 380: Strengthen analysis by presenting SIMBA thermal profile time series

Although we acknowledge that the SIMBA-recorded temperature profiles may be of interest to some readers, as noted in response to Comment 12, we believe that these temperature profiles would not offer additional insight beyond what is already presented in Figure 6. Additionally, we have provided two reference Figures (R1 and R2) in response to this Reviewer's comment which aim to clarify water column stratification dynamics in greater details.

17. Figure 8: Provide model results across BT = 0 to -10°C (not 0 to -5°C) to justify selecting BT = -5°C as optimal. Include sensitivity analysis of BT variations.

Figure 8 provides the optimal results for BT=-5°C while Table 6 provides the optimal parameters. We have provided all model results for BT=0 to -10°C for BUDs, FUDs, and parameters $\alpha$, $a$, $b$, and $C$ in Table A1 which is now incorporated as part of the main text of the paper per your following comment.

Response to Reviewer 1:

18. Lines 387 vs. 395: Conflicting descriptions of BT experimental ranges ("0 to -5°C" vs. "0 to -10°C").

We thank the Reviewer for point out this discrepancy. We have revised L387 so that the BT experimental ranges are consistent (0°C to -10°C).

19. Lines 379–399: Reorganize logic: Step 1: Present BT sensitivity experiments. Step 2: Identify optimal BT (-5°C). Step 3: Report corresponding $\alpha$, $a$, $b$, $c$ Step 4: Show final model performance (Figure 8).

We agree with this comment that the suggested workplace provides better clarity. We have adopted this logic as suggested in the revised manuscript.

20. Lines 411–417: Improve readability by integrating Table A1 into the main text.

We have now integrated Table A1 into the main text, as recommended by this Reviewer.

21. Lines 399–417: Relocate to the Discussion section to critically evaluate: (1) Model applicability across lake types. (2) Limitations in parameter transferability.

This section has been reworked with many parts of L399-417 relocated to the new discussion section as requested. In response to 1), we have made note within the newly created discussion section that the model should only be applied for lakes with similar geometric properties to Landing Lake, *i.e.* small (e.g. < 5 km²) and shallow (e.g. <5 m deep). For 2), we also make note that parameters should be established for each region uniquely but once established using data from one or a few lakes in a particular region, the parameters could be used for regional scales.

22. Lines 461-463: The statement is debatable, as freeboard can be estimated using Archimedes' principle.

We agree that this statement may be debated. It is correct that Archimedes' principle can be used and is in fact most commonly used. The use of Archimedes' principle is still challenging in practice due to accurate estimates of snow densities. If accurate measurements are available, then freeboard can be readily estimated. We have softened the language of this statement to reflect that the statement is debatable.

*The addition of a freeboard component to the model would likely improve simulation strength but at a cost of increased complexity and uncertainty as the accurate estimation of freeboard could be challenging.*

23. While the authors aim to develop a simplified empirical model for ice thickness estimation, the interannual variability of coefficients ($\alpha$, a, b, c) necessitates field-based calibration, severely limiting practical utility. To strengthen conclusions, I recommend: (1) Comparative studies across lakes to establish parameter ranges. (2) Explicit guidance on minimum data requirements (e.g., duration and type of meteorological/hydrological inputs) for reliable model application in the future.

We appreciate the comment by this Reviewer and for the useful criticism. The intention of this study was to establish and present empirical relationships that could be used for simulating ice thicknesses in Landing Lake. While we acknowledge that readers may be interested in variability of the

Response to Reviewer 1:

parameters across the Northwest Territories in lakes of varying sizes, this was not the intended purpose of the presented study. We do however now note in the new discussion section that parameter ranges can be established by following a similar analysis in other watersheds or hydroclimatic regions where time series of air temperatures, snowfall, and a few ice thickness measurements are available.

Here we have applied the empirical relationships in Equations 5b, 5c, and 6 for simulation of ice thicknesses in Vee Lake (62.55113°N, 114.35578°W), a lake located ~2.5 km from Landing Lake. This lake was selected as we have collected frequent in-situ measurements using a SIMBA for the periods of roughly Nov. 8, 2022- Dec. 31, 2022, and Nov. 15, 2024- Dec. 31, 2024, which are used for validation. Vee lake has similar properties as Landing Lake, with a surface area ~0.8 km², 5.80 m max depth, 1.58 m average depth. We also present the empirical $CFDD$ vs. $S_T$ curve (Figure R3) for Oct.-Dec. 2024 using the Yellowknife Airport weather station for reference with derived parameters $a$=51.52, $b$=0.037, and $C$=0.87. An $\alpha = 0.48$ was selected for application 2024 to account for higher $CFDD$ in 2024 as compared to 2023. For 2022, we use $\alpha$=0.44, $a$=2.91, $b$=0.050, and $C$=0.047 (Table 6). Results are presented in Figure R4.

[Figure]

**Figure R3: Cumulative freezing degree days (BT=-5°C) versus cumulative snowfall measured at the Yellowknife Airport Meteorological Station between Oct.-Dec. 2021 to 2024.**

Response to Reviewer 1:

[Figure]

**Figure R4: Application of empirical models to a small (~0.8 km²) and shallow (5.80 m max depth, 1.58 m average depth) lake located ~2.5 km from Landing Lake for 2022 and 2024. Modelled ice thicknesses are compared with in-situ measurements collected by a SIMBA installed in Vee Lake.**

Results show good accuracies in simulated ice thicknesses when compared to in-situ SIMBA measurements (RMSE<= 5.4 cm). FUDs in 2022 for Vee Lake are uncertain as the SIMBA is installed after ice is thick enough to walk. However, based on Sentinel 2 optical imagery, FUDs for Vee Lake are thought to be within a few days following Oct.22, 2022 (but no confirmed date), and between Oct. 22-23 in 2024. Modelled FUDs were Oct. 22, 2022, and Oct. 18, 2024, putting the approximate error in FUDs at 0-5 days. This accuracy would be deemed appropriate as a first-order approximation. From this analysis, it appears that three years worth of data proved sufficient to adequately simulate ice thicknesses in a nearby lake.

24. Section 8 (Lines 488–524): Restructure content: (1) Relocate technical discussions (e.g., model assumptions) to the Discussion section. (2) Retain application scenarios and future research directions in Conclusions

We have restructured the manuscript following this, and previous comments by this Reviewer. We greatly appreciate this Reviewer's time in providing critical feedback and suggestions for improvement and believe we have responded to all suggestions and concerns appropriately.

---

## Author Comment (AC2)

**Reviewer comments are provided in black.**
**Author responses are provided in blue**

Review on "Interannual variability in air temperature and snow drive differences in ice formation and growth" by Arash Rafat and Homa Kheyrollah Pour

Climate change involves many complex processes. For the cryosphere, the timing of the events is a critical concept. The freezing and melting of lake/sea ice alter the energy balance between the atmosphere and the underlying water bodies (lakes or oceans), thereby influencing climate dynamics. One of the most critical practical concerns is that freeze-up timing directly affects the usability of ice roads. This is especially vital for North American Arctic communities, where ice roads serve as lifelines for remote regions in Alaska and Canada.

This manuscript investigates ice formation and growth in a small boreal lake in Canada's Northwest Territories (NWT). The authors conducted *in-situ* observations over three consecutive winter seasons on a single lake. The dataset includes local meteorological parameters such as wind speed, air temperature, and turbulent and radiative heat fluxes. A platform was installed on the lake to collect high-resolution snow and ice temperature measurements using a novel, cost-effective automated device (SIMBA).

These observations, combined with long-term meteorological data from weather stations, were used in a statistical model to calculate ice thickness employing an exponential function of snowfall as input.

The manuscript investigates local variability in climate and weather, particularly ice formation and growth, with a focus on ice freeze-up dates (FUD) and the evolution of snow and ice cover. The authors argue that the derived relationships between air temperature, snow depth, and ice thickness can be used to predict the minimum ice thickness required for ice road construction, aiding in the effective management of construction activities.

The topic of this manuscript is highly relevant to the scope of *TC*. The observations were made without flaws, and the configuration of the SIMBA platform is solid and well-justified. The statistical model is conventional yet robust, and the data analysis is convincing. However, I have some concerns and comments regarding certain aspects of the content, which I hope the authors will address through a proper revision before the manuscript's final acceptance.

**Major comments**

1.  The manuscript's overall structure could be improved for better clarity. a) I don't see a clear chapter on the "Results". The presentation of data and results was somehow mixed. I suggest restructuring the entire manuscript. For example, a chapter entitled "results" that contains partial Chapter 4 and Chapters 5 and 6 may yield better clarity of the manuscript; b) I am not sure why Chapter 8 is needed, especially after the conclusions have been made. I suggest this chapter can be placed before the conclusion, e.g., Discussions.

    We thank this Reviewer for this comment and have revised the manuscript to now include clear results and discussion sections. Chapter 8 now forms part of the discussion section as recommended.

2. Based on the study's objective, as stated in the abstract and final chapter of this manuscript, the discovery of robust relationships between air temperatures, snow cover, and ice thickness is intended to assess the feasibility of ice road construction and support effective construction management, which I agree. However, this work has been carried out in a tiny lake (1.1 km2). The questions I want to ask: a) How representative are the results of this work? b) Can those derived formulae be applied to obtain FUSs in other parts of the NWT? c) Would it be possible to assess the performance of the formula for the other small lakes in NWT?  d) At least a discussion of the general applicability of the formula should be included in this study.

We thank this Reviewer for this comment and acknowledge a similar comment by Reviewer 1. For a) the results of this work are highly representative for many regions of the NWT which are scattered with thousands of small (*e.gs.* < 5 km²) and shallow lakes (*e.gs.* <10 m) which are commonly used for transportation, recreation, and cultural activities.  For b), the applied formulae can be used to approximate the FUD for other parts of the NWT and more broadly, Northern Canada; however, some caution should be applied. FUDs depend highly on lake volume and as such, we don't not recommend applying these equations for large and deep lakes. Instead, these equations can be useful for lakes with similar properties to Landing Lake. Some examples may include typical thermokarst lakes formed in shallow ground ice settings which are widespread across circumpolar regions and have relatively shallow depth (*e.g.* <10 m; West and Plug, 2008; Bouchard et al. 2016).

For ice road construction, accurate ice thickness prediction would be top of mind, as construction activities would necessitate a minimum ice thickness to be reached prior to snow clearing for the ice road. For this purpose, appropriate parameter ranges would need to be applied and developed for different regions where snowfall, air temperatures, and a few measurements of ice thickness are available. The former two measurements would be used to derive parameters $a$ and $b$ while the later used to derive $\alpha$. Here, we develop these parameter ranges for one region (Yellowknife), but relationships could be developed for any region where appropriate data exists.

We would like answer c) with applying the empirical relationships for a lake located ~2.5 km from Landing Lake (Vee Lake: 62.55113°N, 114.35578°W). Vee Lake has similar properties as Landing Lake, with a surface area ~0.8 km², 5.80 m max depth, 1.58 m average depth. This lake was selected as we have in-situ measurements collected by a SIMBA for the periods of roughly Nov. 8, 2022- Dec. 31, 2022, and Nov. 15, 2024- Dec. 31, 2024 for use in validation of the empirical model. Equations are provided below for reference:

$$CFDD = ae^{bS_T} \tag{5b}$$

$$h_i = \alpha \left( \sqrt{\frac{2k_i}{\rho_i L}} \right) ae^{0.5bS_T} \tag{5c}$$

$$h_i = Ce^{0.5bS_T} \tag{6}$$

Given the proximity to Landing Lake and the Yellowknife Airport meteorological station, we can apply the same parameters developed from curves of $CFDD$ vs $S_T$ that were used for Landing Lake in 2022. We have also derived parameter values for Oct.-Dec. 2024 using the Yellowknife Airport weather station and provide a reference Figure (Figure R3) below.  For 2024, derived parameters are $a$=51.52, $b$=0.037, and $C$=0.87. An $\alpha$ = 0.48 was selected for application 2024 to account for higher $CFDD$ in 2024 as compared to 2023. For 2022, we use $\alpha$=0.44, $a$=2.91, $b$=0.050, and $C$=0.047 for $k_i$=2.2 W m$^{-1}$ °C$^{-1}$, $L$= 334000 J kg$^{-1}$, $\rho_i$=916 kg m$^{-3}$. These values were identical to those used for Landing Lake in Table 6 for 2022. Results are presented in Figure R4.

Based on this curve, and with higher *CFDD* as compared to 2023, we choose $\alpha$ to 0.48. This is ~10% higher than in $\alpha$ in 2023 to account for the higher *CFDD*. Parameters used can be summarized as follows: In 2022: we use $\alpha$=0.44, $a$=2.91, $b$=0.050, and $C$=0.047 (Table 6), and for 2024: $\alpha$=0.48, $a$=51.52, $b$=0.037, and $C$=0.87.

[Figure]

**Figure R3: Cumulative freezing degree days (BT=-5°C) versus cumulative snowfall measured at the Yellowknife Airport Meteorological Station between Oct.-Dec. 2021 to 2024.**

[Figure]

**Figure R4: Application of empirical models to a small (~0.8 km²) and shallow (5.80 m max depth, 1.58 m average depth) lake located ~2.5 km from Landing Lake for 2022 and 2024. Modelled ice thicknesses are compared with in-situ measurements collected by a SIMBA installed in Vee Lake.**

Results show good ice thickness simulation accuracies (RMSE<= 5.4 cm). FUDs in 2022 for Vee Lake are uncertain as the SIMBA is installed after ice is thick enough to walk. However, based on Sentinel 2 optical imagery, FUDs for Vee Lake are thought to be within a few days following Oct.22 2022 (but no confirmed date), and between Oct. 22-23 in 2024. Modelled FUDs were Oct. 22, 2022, and Oct. 18, 2024, putting the approximate error in FUDs at 0-5 days. This accuracy would be deemed appropriate as a first-order approximation.

Following comment d), we have added in a small discussion on the applicability of this approach and some limitations of the model to the new discussion section of the paper. We hope that the presented analysis adequately answers your concerns. This discussion is presented below:

*The presented empirical models demonstrate an effective means of simulating total ice and snow ice thicknesses in Landing Lake using snowfall and air temperatures recorded from the Yellowknife Airport weather station, located 11 km south of Landing Lake. Relationships between CFDD and $S_T$ (Fig. 8) can be considered as regional relationships which can be applied to other Yellowknife-area small lakes with similar lake depths (e.g. < 5 m) and surface areas (e.g. < 5 km²) for first order estimates of ice thicknesses. Further, the same methodology can be applied for establishing values of $\alpha$ $\gamma$, $b$, $C$, $D_1$, $D_2$, and $D_3$ in other regions of the Northwest Territories if measurements of snowfall, air temperatures, and a few measurements of ice thickness are available. This analysis is not intended for use in large and deep lakes whose latency effects during freeze-up would require unique treatment. Multi-year monitoring in other regions of the Northwest Territories can aid in establishing regional curves such as those presented in Fig. 8 for determining inter-annual and regional variability in model parameters.*

References:

Bouchard, F., MacDonald, L. A., Turner, K.W., Thienpont, J.R., Medeiros, A.S., Biskaborn, B.K., Korosi, J., Hall, R.I., Pienitz, R. and Wolfe, B.B. 2017. Paleolimnology of thermokarst lakes: a window into permafrost landscape evolution. Arctic Science. 3(2): 91-117. https://doi.org/10.1139/as-2016-0022

West, J.J. and Plug, L.J. 2008. Time-dependent morphology of thaw lakes and taliks in deep and shallow ground ice. Journal of Geophysical Research Earth Surface, 113, F01009

3.  I have problems understanding the presentation of figures and tables. Many captions are currently insufficient for readers to easily grasp the key information. I recommend revising them accordingly (see my detailed comments below).

    Thanks for the Reviewer's comment. We have expanded on and clarified as suggested in your detailed comments.

4.  Authors investigated several snow parameters: date of the first snowfall ($S_{ON}$), the cumulative snowfall ($S_T$), the peak hourly snowfall rate in a given day in each month ($S_p$), and the number of snowfall days ($S_d$). Please explain a bit more about $S_p$. Based on the definition, I understand the

other parameters are one number for each winter season. However, $S_p$ has multiple numbers for each winter, right?

Thank you for your inquiry. Parameter $S_p$ represents the peak hourly snowfall rate recorded in each day over a given month and therefore has one value for each month per winter season. Here, the analysis is done from Oct.-Dec. so each season will have 4 values. We present these values in Table 2. For use in the frequency analysis (Figure 4), the maximum $S_p$ value for each season is used. Parameters $S_{ON}$, $S_T$, and $S_d$ can represent only one value that is cumulative of the study period (Oct.-Dec.). For instance, $S_d$ can represent that total number of snowfall days between Oct.-Dec. This approach of having one number for each winter was used for the frequency analysis (Figure 4). However, $S_d$ and $S_T$ can also be interpreted for each month to better understand within month weather and climate variability (e.g. Table 1 and Table 2). $S_{ON}$ always has one value as it represents the date of first snowfall for each winter.

The snow measurement was made in the "Yellowknife Airport weather station (1942-2023)". Please write more information about snow observations, e.g., instrumentation, data quality, and possible errors.

Thank you for the inquiry. Snowfall data was collected since 1943 using manually methods and automated methods. Manual methods include measuring the amount of freshly fallen snow on a snowboard- this was the approach used in many Canadian weather stations (see Fischer, 2011). The exact date of when automated methods replaced manual methods is currently unknown; however, appropriate inquiries have been made to ascertain these dates. Automated measurements of snowfall are commonly collected using an SR50 Ultrasonic Snow Depth Sensor. Several challenges have been noted in the past on measuring snowfall/depth using SR50s including penetration of ultrasonic pulses through fresh snowfall layers and reflection by deeper, denser layers (Goodison et al. 1988), and lack of spatial representativeness (Fischer, 2011). Data quality assurance and control is provided by ECCC. Data quality flags are described in ECCC's technical documentation here: https://climate.weather.gc.ca/doc/Technical_Documentation.pdf

Fischer, A. 2011. The Measurement Factors in Estimating Snowfall Derived from Snow Cover Surfaces Using Acoustic Snow Depth Sensors. *Journal of Applied Meteorology and Climatology*, 50(3), 681-699. https://doi.org/10.1175/2010JAMC2408.1

Goodison, B. E., Metcalfe, J. R. and Wilson, R. A. 1988. Development and performance of a Canadian automatic snow depth sensor. WMO Instruments and Observing Methods Rep. 33, 317–320.

If there are good snowfall data, I encourage authors to apply an analytical model to calculate the ice thickness and snow-ice, taking into account the effect of snow. See Lepparanta (1993) for a good example of such an analytical model.

This is a great suggestion. An analytical approach can readily be used to estimate total ice thickness. In this study, we intended to only present a statistical approach based on measured data to estimate ice thicknesses but appreciate the applicability of some analytical or semi-analytical methods. As suggested, we applied Equation (20) from Leppäranta (1993): $H^2 = \frac{2k_i S}{[p_i L\left(1+\frac{\lambda k_i}{k_s}\right)]}$, to measured ice thicknesses and present results here. Applied parameters included $p_i$=916 kg m$^{-3}$, $k_i$= 2.2 W m$^{-1}$ °C$^{-1}$, $k_s$= 0.1 W m$^{-1}$ °C$^{-1}$, and $L$= 334 000 J kg$^{-1}$. We note to the reader that $\lambda$ refers to the ratio of snow depths to ice thicknesses, commonly >0.5. Here, we provide solutions using

Equation (20) using both SIMBA measured values of $\lambda$ averaged over the season (Figure R5, d, e, and f), and also for optimized values (Figure R5, a, b, and c). Optimized values used $\lambda$ as a calibration parameter to reduce mean absolute errors between SIMBA measured ice thicknesses and those modelled by Equation (20).

[Figure]

**Figure R5:** Comparison of SIMBA measured ice thicknesses with an analytical model for ice growth presented in Leppäranta (1993). Plots a, b, and c present results using optimized values of parameter $\lambda$, while plots d, e, and f present solutions with average $\lambda$ calculated using SIMBA measured snow and ice thicknesses.

We observe that errors remain relatively low (less than 5 cm) when $\lambda$ is calibrated, however, errors increase significantly when using $\lambda$ measured from field data.

In this study, we chose not to apply an analytical model for snow-ice formation, as we do not present detailed snow density measurement. Without such data, implementing an analytical model would be challenging and would rely heavily on parameter calibration to achieve desired and meaningful results.

*Leppäranta M. 1993. A review of analytical models of sea-ice growth. Atmosphere-Ocean, 31(1): 123-138, doi:10.1080/07055900.1993. 9649465.*

**Detailed comments:**

5.  Please add a Canada map as a background for Figure 1. I think Photo A can be dropped since Figure 2 shows the details of the SIMBA floating station.

We thank the Reviewer for this helpful suggestion. We explored incorporating a full Canada map as a background for Figure 1 but found that doing so significantly reduced the clarity and spatial

detail of the Landing Lake region. Instead, we included a smaller reference map of Canada in the lower left-hand corner of Figure 1 to maintain geographic context.

Regarding Photo A, while we recognize that Figure 2 provides a detailed conceptual overview of the SIMBA Floating Research Station, we believe that Photo A remains valuable as it offers a real-world visual reference of the station's physical setup. In our view, this complements the schematic illustration in Figure 2 and enhances the reader's overall understanding.

6. "Photographs 1, 2, and 3 were taken on October 23, 2023, 105 at 09:00 local time."  I don't see any close text to explain those photos.  I found at L290, a description "culminating in a FUD of October 23, 2022, 3 days", I would assume this was the explanation of those photos. If so, maybe write: ,,,at 09:00 local time (see explanation in 5.1), and correct the typo 2022 to 2023. Otherwise, please add text somewhere to explain those photos.

We thank the Reviewer for noting this. The trail camera photographs in Figure 1 were mainly shown to provide the reader with a perspective few of ice conditions around different sections of the lake during freeze-up. We have extended the caption to describe the intention behind these photographs:

*Figure 1: Site map of Landing Lake, Northwest Territories, including photographs of the Floating Research Station, and perspectives from trail cameras (1, 2, and 3). Photographs 1, 2, and 3 were taken on October 23, 2023, at 09:00 local time and present three perspectives of the lake during the freezing process.*

7. Could you edit photo 3 in figure 1 to show a horizontal coastal line?

We have rotated photo 3 in Figure 1 as per your request so that the coast line is horizontal.

8. It seems to me that the PAR and pressure transducers' data on FRS, as well as turbulent and radiative heat fluxes measurement at the weather station on land nearby, are not used in this study. Please include a brief description of the purpose of these data.

The primary objective of presenting these measurements was to showcase the full instrumentation setup used at Landing Lake, including the meteorological station on the island and additional sensors on the FRS platform. We acknowledge that not all the collected data were utilized in the present analysis. The turbulent and radiative heat flux data are essential for characterizing the lake's complete annual energy balance. These fluxes are measured at a comprehensive meteorological station located on an island in Landing Lake, while supplementary instrumentation for shortwave radiation measurements is installed on the FRS (Figure 2). Additionally, PAR sensors within the water column are used to estimate light extinction during both the ice-covered and open-water seasons. Although these datasets are not used in this study, they form an integral part of our long-term research framework. Some portions of the data have been published in earlier works (Rafat et al., 2023, 2024; Rafat and Kheyrollah Pour, 2023), and we intend to further explore and publish the remaining datasets in future studies.

9. Figure 2. Please add "lake water surface or ice surface close to the black inverted triangle symbol.

The terms "water surface" have been added to Figure 2 where requested.

10. Section 3.3 Heat storage: please explain the physical meaning of negative heat storage in this section. Such values were calculated around L315.

We thank the Reviewer for this inquiry. The heat storage here is calculated as a rate of change of mean lake heat content over a one hour period relative to the previous hour. Therefore, a negative number would indicate a cooling or loss of heat in the lake while a positive value indicated.

11. Figure 3. Please explain the symbol "x". It is hard to see the yellow color of the triangle.

The 'x' in Figure 3 indicates extreme values that were >±2.7 standard deviations away from the mean data. We agree that the yellow triangles are difficult to see and have therefore re-rendered the figure with a new colour, provided below for reference.

[Figure]

**Figure 2: Comparison of a) cumulative monthly snowfall and b) mean daily air temperatures for the September-December period in 2021, 2022, and 2023 against the climate normal (1981-2010) and preceding 30-year record (1992-2021) periods.**

12. L219: "The same year saw colder than normal conditions by the end of December with *CFDD* being 123% of normal". Please add the number before 123%. Also for those >100% in the following text until L230 if possible.

We have added in the numbers (magnitudes) of snowfall for the percentages as per your recommendation.

13. Table 1. Please explain what those numbers with parentheses (4.1, 10.5). In the Table, I see (Tmin - Tmax),*mean (min, max) monthly cumulative snowfall between 1981-2010 and 1992-2021.

The bracketed values represent the minimum and maximum daily mean air temperatures in each month. For snowfall, bracketed values represent the minimum and maximum cumulative snowfall values recorded in a given month within the climate normal 1981-2010 and 1992-2021 periods. We see the possible confusion in Table 1 as the dates for the climate normal period "(1981-2010)" are also in parentheses. We have removed parentheses around "(1981-2010)" to avoid this confusion.

14. I would remote "-" on the 3rd line. Please explain a bit more Sp, see my major comment 4.

"-" from the third line in Table 2 has been removed as per your request. Sp has been further explained further in the response to major comment 4.

15. Figure 5. Please explain how the surface temperature was measured by SIMBA. Could you add air temperature measurements from the nearby weather station on land for comparison?

Surface temperatures in 2022 and 2023 were measured directly by the SIMBA by noting the temperature reading at the identified air-water interface along the SIMBA thermistor chain. We have added in this information to the caption. Measurements of air temperatures from the Yellowknife Airport weather station compare well to those recorded by the SIMBAs at Landing Lake. Please see a comparison below in Figure R6 of daily mean air temperatures. Values are nearly identical except for slight deviations in very cold weather. We have opted not to include the Ta measured from the Airport to Figure 5 as it would make interpretation difficult due to line clutter.

[Figure]

**Figure R6: Comparison of daily mean air temperatures measured by the SIMBA and at the Yellowknife Airport weather station located ~11 km away.**

16. Table 5. What is "X"? maybe N/A is better?

We agree that "N/A" is clearer. This has been changed.

17. Figure 8. Please explain how those dots (blue, black and red) were obtained.

The dots represent values of $CFDD$ and $S_T$ recorded at the Yellowknife weather station. This has been added to the caption for clarity.

18. Figure 11. Please explain how the observed snow-ice thicknesses were made?

We acknowledge that no clear description was provided in the methods section for how snow-ice was calculated. Rather than inserting how these measurements were made within Figure 11's caption, we chose to add in a description within Section 3.1: Interface detection and manual observations.

*Ice thicknesses were calculated as the difference between the identified ice bottom and surfaces, while snow depths calculated as the difference between the snow and ice surfaces. If the ice surface is identified at a position that is higher up the chain than its original position, snow-ice has formed. Therefore, snow-ice thicknesses could be calculated as the difference in these positions.*

19. L406: CFDD should b*e CFDD.*

This has been corrected.

20. L79: "inter- and intra- annual variability", maybe good to write "inter-annual and seasonal variability".

We have adopted this phrasing as suggested.

21. "Code and data availability". I am not sure whether the statement the authors made is acceptable to the TC. The data link (https://climate.weather.gc.ca/) is the main page of ECCC. I think the authors should provide the data link that can direct access the air temperatures and snowfall measurements at the Yellowknife Airport weather station between 1942-2023. The lake measurement data sets (weather station, SIMBA) were missing and should be publicly accessible.

We thank the Reviewer for this note.

As recommended, we will provide direct access to the air temperature and snowfall data from the Yellowknife Airport weather station (1942–2023), rather than linking to the general homepage of the ECCC website. We have combined this data into an csv file for ease of use by the public.

We appreciate all the comments provided by this Reviewer and would like to thank the Reviewer for their time in critically assessing the contents of this manuscript.

---

## Author Response (AR2)

**Response to Reviewer Reports**

**Reviewer comments are provided in black.**
**Author responses are provided in blue**

**Reviewer 1**

The authors have substantially revised the manuscript, and I find the revised version satisfactory. However, I noticed that another reviewer also raised concerns about the representativeness of the study area. To further highlight this aspect, the authors could consider processing the HydroLAKES dataset (Messager et al., 2016) to provide statistics on the number of small (e.g., <5 km²) and shallow (e.g., <10 m) lakes in Canada's Northwest Territories (NWT), thereby emphasizing the prevalence and significance of such lakes. Including this information in the introduction would strengthen the manuscript. Aside from this minor point, I have no further comments.

Messager, M. L., Lehner, B., Grill, G., Nedeva, I., & Schmitt, O. (2016). Estimating the volume and age of water stored in global lakes using a geo-statistical approach. Nature Communications, 7(1), 13603. https://doi.org/10.1038/ncomms13603

We thank the Reviewer for the constructive feedback throughout the peer-review process. As recommended in this Reviewer's Report, we processed the HydroLAKES dataset and identified ~138,000 small (< 5 km$^2$) and shallow (<5m mean depth) lakes and ponds within the border of the Northwest Territories (NWT), Canada with a minimum size of 0.1 km$^2$. This accounted for ~86% of all lakes and ponds within the NWT contained within the HydroLAKES dataset. We have added this information to the manuscript.

**Reviewer 2**

I see authors have made a major revision of the manuscript. Authors have responded adequately to the comments from both reviewers. I am satisfied with the rebuttal letter and the revised manuscript. I don't have further scientific comments to the manuscript. I think authors still need to comply with the TC data policy. However, this is a technical issue, and I leave it to the handling editor to discuss with the authors. Otherwise, I am pleased to recommend this manuscript for publication in TC. Congratulations!

We appreciate the comments of this Reviewer throughout the review process. We have amended the Code and data availability statement to include a published dataset in an appropriate repository to comply with the data policy. https://doi.org/10.5683/SP3/QZJVYD